# Evolution of sexual conflict in scorpionflies

**Agnieszka Soszyńska-Maj[1]\*, Ewa Krzemińska[2]\*, Ricardo Pérez-de la Fuente[3], Ji-Shen Wang[4], Krzysztof Szpila[5], Kornelia Skibińska[2], Katarzyna Kopeć[2], Wiesław Krzemiński[2]**

[1]Faculty of Biology and Environmental Protection, Department of Invertebrate Zoology and Hydrobiology, University of Łódź, Łódź, Poland; [2]Institute of Systematics and Evolution of Animals, Polish Academy of Sciences, Kraków, Poland; [3]Oxford University Museum of Natural History, Oxford, United Kingdom; [4]College of Agriculture and Biological Sciences, Dali University, Dali, China; [5]Department of Ecology and Biogeography, Faculty of Biological and Veterinary Sciences, Nicolaus Copernicus University, Toruń, Poland

**Abstract** Sexual conflict – opposite reproductive/genetic interests between sexes – can be a significant driver of insect evolution. Scorpionflies (Insecta: Mecoptera) are models in sexual conflict research due to their large variety of mating practices, including coercive behaviour and nuptial gift provisioning. However, the role of palaeontology in sexual conflict studies remains negligible, namely due to the paucity of well-preserved fossils. Here, we describe three male scorpionflies from Cretaceous and Eocene ambers. The structure of notal and postnotal organs is analysed in extant and extinct forms; a depression below the base of the notal organ in different panorpid species spatially matches the anterior fold of the female's wing. Based on disparate abdominal configurations and correlations in extant relatives, we posit that each new fossil taxon had a different mating approach along a nuptial gifting-coercive spectrum. The Eocene specimen possesses extreme female clamping abdominal armature, suggesting a degree of sexual coercion greater than in any other known scorpionfly, extinct or extant. The fossil record of abdominal modifications in male scorpionflies documents a relatively late evolution (Eocene) of long notal organs indicating oppressive behaviour toward a female during mating. Our findings reveal a wider array of mating-related morphological specialisations among extinct Panorpoidea, likely reflecting more diversified past mating strategies and behaviours in this group, and represent first steps towards gaining a deep-time perspective on the evolution of sexual conflict over mating among insects.

\*For correspondence:
agnieszka.soszynska@biol.uni.
lodz.pl (AS-M);
ekrzeminska9@gmail.com (EK)

**Competing interest:** The authors declare that no competing interests exist.

## Editor's evaluation

The authors present, describe and interpret exquisite insect fossils preserved in amber. The detailed scorpionfly morphology reveals specific reproductive morphological adaptations that are clear enough to corroborate hypotheses explaining how these insects may have mated. Based on careful morphological comparisons with extant scorpionfly species, the authors are able to speculate on how sexual conflict evolved and shaped insect mating behavior. The findings are of interest to evolutionary biologists studying reproduction and behavior, entomologists and paleontologists. Given the unusual diversity in insect mating behavior, which is simultaneously fascinating and horrifying, this deep-time perspective is likely to interest not only colleagues in the field, but also the general public.

## Introduction

Sexual conflict between two conspecific sexes arises when divergent interests about the timing and duration of copulation, number of mating partners, or parental investment occur, provided that a trait enhancing the reproductive success (~fitness) of one sex may reduce that of the other (**Arnqvist and Rowe, 2005**). Currently more broadly defined as the sexually antagonistic selection of shared traits, the sexual conflict is now recognised as a "pervasive evolutionary selective force", leading in many known instances to a male-female coevolution (**Parker, 1979**; **Rostant et al., 2020**). At the very core of this conflict lies the struggle for dominance over genetic control in offspring by each sex (**Thornhill, 1976**; **Thornhill, 1979**). The study of mating practices offers one of the most direct ways to address sexual conflict in nature. Two seemingly opposite (yet not necessarily mutually exclusive) mating-related strategies are observed in males: aggressive behaviour toward a female, which allows exerting domination while mating, and the nuptial gifting of food to allure and/or pacify a female during the time of copulation and sperm transfer (**Lehmann, 2012**; **McCartney et al., 2013**). Both coercive and nuptial gifting strategies appear costly to males because they are based on the differential allocation of limited resources (**Liu et al., 2015**); therefore, an exaggerated armature may hinder the overall fitness in a male (e.g. in rhinoceros and flour beetles a larger 'weapon' is negatively correlated with wing size; **Yamane et al., 2010**). In order to incline the female to mate, males show aggression (e.g. calopterygid damselflies; **Rivera and Andrés, 2002**), intimidating behaviour (e.g. water striders; **Han and Jablonski, 2010**), and may use special structures, such as modified front legs to clasp the female wings (e.g. sepsid flies; **Eberhard, 2002**) or adhesive discs to secure grip on the female (e.g. diving beetles; **Bergsten et al., 2001**). Within coercive mating, extreme cases of aggressive behavior by males, which may cause damage or even death to the female, are known among insects and other invertebrate groups, such as true bugs (**Siva-Jothy, 2006**), water bugs, and camel-spiders (**Arnqvist and Rowe, 2002**; **Hrušková-Martišová et al., 2010**) or snails (**Kimura and Chiba, 2015**), but also among vertebrates (e.g. ducks; **Brennan and Prum, 2012**). Nuptial gifting is also widespread. Crickets and bush crickets (Orthoptera: Tettigonidea, Grylloidea) provide an edible gift attached to a structure containing the sperms. Female consume the nutritious part during fertilisation by the spermatozoans, the time for consumption correlating with that of fertilisation (**Lehmann, 2012**; **McCartney et al., 2013**). Males of other orthopterans offer thoracic secretions (Gryllidae; **Bussière et al., 2005**) or even their own hind wings as nuptial gifts (Haglidae; **Eggert and Sakaluk, 1994**; **Sakaluk et al., 2004**). Silk-wrapped nuptial gifts of prey are offered by spiders (**Stålhandske, 2001**) and dance flies (Diptera: Empididae; **Preston-Mafham, 1999**; **Sadowski et al., 1999**; **Lebas and Hockham, 2005**); the shining silk cover makes a gift more enticing and may deceive the female if it only contains a scarce – or even fully absent – edible portion (**Ghislandi et al., 2017**). In dobson flies (Megaloptera: Corydalidae), the species offering nuptial gifts have a 'weapon' armature used for fighting off the rivals smaller than in species whose males rely only (or mostly) on such armature (**Liu et al., 2015**). Among mecopterans, a gift of prey is offered by hangingflies (Bittacidae; **Thornhill, 1976**) and scorpionflies (*Panorpa* spp.; **Gwynne, 1984**); the latter may offer hardened salivary masses to the female instead (**Thornhill, 1979**; **Bockwinkel and Sauer, 1994**). Coercive and nuptial-gifting mating approaches can co-occur in same species, which then exhibits a mixed strategy that can serve as a model to study the trade-off between these two tactics (e.g. **Tong and Hua, 2019**).

Scorpionflies (Mecoptera: Panorpidae) are models in sexual conflict research. Their mating practices range from coercive to provisioning nuptial gifts, in extreme cases including mouth-to-mouth salivary transfer until the end of copulation (**Zhong et al., 2015**). Mecoptera are among the oldest holometabolous insects. They are known since the Permian, when they were a significant component of the flying insect fauna. Whereas scorpionflies flourished in the Mesozoic, most of their lineages rapidly declined during the Cenozoic. In fact, mecopterans are considered to have gone through a degree of extinction unparalleled among holometabolans (**Grimaldi and Engel, 2005**). Extant Mecoptera are classified in nine families, most of which are relict; only two lineages, Panorpidae and Bittacidae, are relatively species-rich. The Panorpidae appeared in the Late Jurassic (**Ding et al., 2014**) and currently represent the most diverse mecopteran lineage, with about 480 described species classified in seven genera (**Bicha, 2018**; **Miao et al., 2019**), that is, c. 67% of extant mecopteran species.

In scorpionfies, the male behaviour towards the female may be coercive in such a degree that the expression 'rape' has been used in the past for describing their mating rituals (**Thornhill, 1980**). In any case, the role of a female in controlling mating and the eventual reproductive outcome appears more

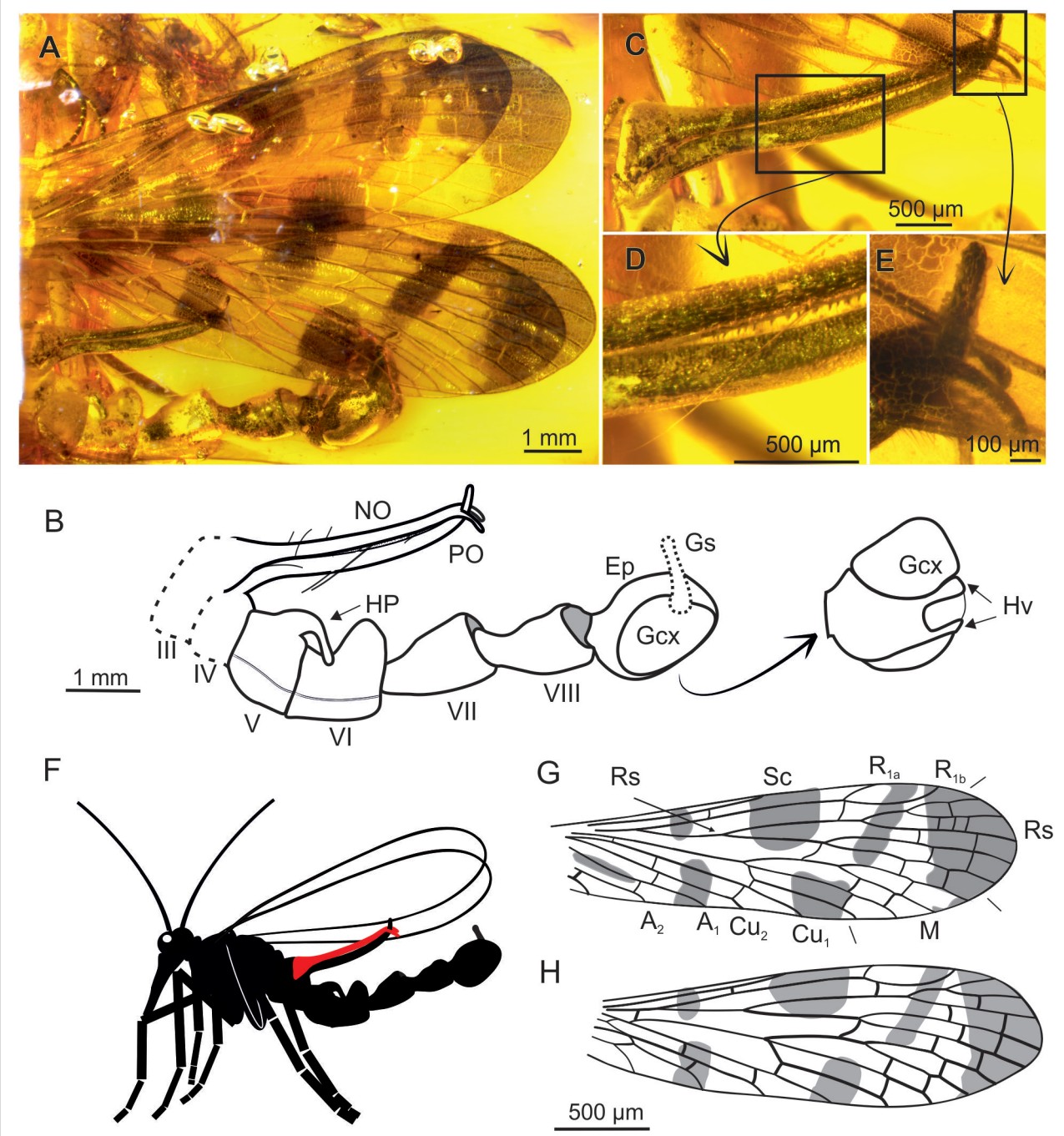

**Figure 1.** *Baltipanorpa oppressiva* sp. nov. (Panorpidae) from Baltic amber (middle Eocene), holotype MP/2711. **A-B**, habitus, photograph of preserved portion (**A**) and explanatory drawing (**B**), with genital bulb in ventrolateral view magnified; **C-E**, notal organ: fully-clasped clamp formed by notal and postnotal organs (**C**), teeth inside the notal-postnotal clamp (**D**), detail of terminal clasp (**E**); **F**, life reconstruction of entire specimen (head and thorax, missing in this species, based on habitus of congeneric *Baltipanorpa damzeni*); **G**, forewing; **H**, hind wing (same scale in both wings). Abbreviations: Ep, epandrium; HP, horn-like process; Hv, hypovalve; NO, notal process; PO, postnotal process; Gs, gonostylus; Gcx, gonocoxite. Red colour: notal process.

active and decisive than initially thought in the group, namely through precopulatory male choice, struggling behaviour to interrupt genital connection, and the selective reception of sperm from gift-providing males (likely through spermathecal muscular action; *Kock et al., 2009*). The modifications of panorpid males enabling coercive mating behaviour include a diverse array of structures on different abdominal segments as well as thinned and elongate last segments of the abdomen. Some of these structures, the so-called notal and postnotal organs, effectively act as a 'clamp', grasping the female's

wing prior to and during the course of copulation (*Figure 1A, B, F*). Depending on the size and shape of these organs in a species, the grasp is more or less coercive. Elongate abdominal segments enable the males to bend the abdomen over that of the female, while additional, usually paired abdominal structures referred to as anal horns are used to achieve and maintain the strained, twisted copulatory position of both abdomens for the long time required for sperm transmission, which could be more than 3 hr (*Tong et al., 2018*; *Tong and Hua, 2019*). Panorpid species with very long notal organs, capable of clamping the entire width of the female wing, only use coercion, so no food gifts are offered (*Tong and Hua, 2019*). In species with mixed strategies, it is the physical condition of the male which determines the followed strategy (*Engqvist and Sauer, 2001*): older and weaker males, as well as smaller ones, primarily adopt an aggressive strategy toward a female, as they are either not able to obtain a prey as a gift or to fabricate it through salivary production, or their gift is often been stolen by a stronger male. Larger males also have an advantage over smaller ones when competing for dead arthropods that can be offered to a female or, if eaten by the male, allow to produce salivary masses. On the other hand, conspecific smaller males bear proportionally larger notal organs (*Thornhill, 1990*), which shows the genetic background of this negative allometry.

Here, we describe three exceptional scorpionflies from three different Cretaceous and Eocene ambers deposits and each belonging to a distinct panorpoid lineage. The disparate abdominal configurations and specialisations in the new taxa allow us to extrapolate trends observed among extant relatives, discussing possible mating habits for the new fossils and previously described ones with preserved non-genitalic abdominal modifications (*Willmann and Novokschonov, 1998*; *Krzemiński and Soszyńska-Maj, 2012*; *Archibald, 2013*; *Wang and Hua, 2020*; *Zhang et al., 2021*). Additionally, the new taxa are of general evolutionary significance, including the description of a new extinct panorpoid family and the first orthophlebiid species described in amber.

## Results

### Systematic palaeontology

Mecoptera *Packard, 1886*
Panorpoidea *Latreille, 1805*
Panorpidae *Latreille, 1805*

*Baltipanorpa* Krzemiński and Soszyńska-Maj, 2012

**Type species.** *Baltipanorpa damzeni* Krzemiński and Soszyńska-Maj, 2012; Baltic amber.
**Other species included.** *Baltipanorpa oppressiva* sp. nov.; Baltic amber.

*Baltipanorpa oppressiva* Soszyńska-Maj and Krzemiński sp. nov.
(Figure 1A–H)
*Zoobank registration:* urn:lsid:zoobank.org:act:F95F5F6C-E210-4CCA-9AAA-2AAB51588C78
**Etymology.** After '*oppressio*', meaning 'coercive' in Latin, referring to the most restraining notal organ ever found within Panorpidae. Gender is feminine.
**Material.** Holotype MP/2711, most part of the abdomen and wings preserved.
**Locality and age.** Baltic amber, middle Eocene (ca. 45 Ma).
**Diagnosis.** Wing membrane with dark maculations (vs. transparent in *B. damzeni*); notal and postnotal processes very long, reaching end of VII segment, notal process distally forked (vs. notal process shorter than postnotal, and not forked in *B. damzeni*); abdominal segment V slightly wider than long; tergite V medially bearing large, strongly recurved process (vs. tergite V without process in *B. damzeni*); segments VII and VIII thinned at base, not particularly elongate, about 3–4× as long as maximal width (vs. segments VII and VIII strongly elongate in *B. damzeni*).
**Description.** Body length 7.2 mm as preserved (only abdomen). *Head.* Antennae partly preserved with 14 cylindrical flagellomeres twice longer than wide. *Wings* (*Figure 1A*). Elongate and narrow, almost 3.5× longer than wide, at least 10 mm long, ~ 3 mm wide. *Forewing* (*Figure 1G*). Membrane bearing seven maculated areas, largest occupying the wing tip; Sc short and single reaching wing margin in midwing, opposite crossvein m-cu; costal area narrow; $R_{1a}$ and $R_{1b}$ surrounding pterostigmal area; Rs with five branches reaching wing margin, $Rs_{1+2}$ longer than $R_{3+4}$; fork of Mb more distal than fork of Rs, crossvein $r_{3+4}$–$m_{1+2}$ sinuous; M with four branches reaching wing margin, $M_{1+2}$ 13× longer than $M_{3+4}$, $M_4$ sharply curved at m-cu; $Cu_1$ and $Cu_2$ fused at base, with two crossveins between them;

three anal veins, $A_1$ reaching wing margin proximal to origin of Rs. *Hind wing* (**Figure 1H**). Venation very similar to forewing, except for: Mb fused with $Cu_1$, $A_1$ proximally fused with $Cu_2$, only one cu2-a2. *Abdomen* (**Figure 1A and B**). Incomplete, longer than wings; first two segments are missing; notal and postnotal processes on tergites III and IV, respectively (**Figure 1B**), both processes very long (1.2 mm), about equal in length, reaching segment VII; notal process with spine-like setae on inner surface's distal half (**Figure 1B and C**), bearing terminal fork curved posteriorly (**Figure 1E**); postnotal process terminally curved anteriorly, lodged between prongs of notal process (**Figure 1B and C**), bearing one very long seta at midlength of organ, and few shorter setae in basal part (**Figure 1C and D**); postnotal organ covered with short and blunt teeth, irregularly distributed on inner surface; tergite V medially bearing a large, strongly recurved process (horn, 300 μm); abdominal segments VI–VIII moderately elongate, with fused tergites and sternites; segments VII–VIII narrower basally. *Genitalia* (**Figure 1B**). Genital bulb round; hypovalves narrow, not extending to apex of gonocoxite; gonostylus shorter than gonocoxite.

## Orthophlebiidae *Handlirsch, 1906*

*Burmorthophlebia* Soszyńska-Maj, Krzemiński and Wang gen. nov.
*Zoobank registration:* urn:lsid:zoobank.org:act:C6F6028E-4A2B-45D2-89EB-4C6736974E6D
**Etymology**. Combination of *Burmo-*, after Burmese amber, and –*orthophlebia*, after the genus *Orthophlebia* (from the Greek 'phlebas', $\phi\lambda\acute\epsilon\beta\alpha\varsigma$, meaning 'veins') to emphasize an alleged close relationship to this genus. Gender is feminine.
**Type species**. *Burmorthophlebia multiprocessa* sp. nov., by monotypy and present designation.
**Diagnosis**. Rostrum long and narrow; antenna with at least 30 flagellomeres. Fore- and hind wings with $R_1$ slightly curved in pterostigmal area towards anterior wing margin; Rs with five branches (vs. six and more in other genera of Orthophlebiidae), $Rs_{1+2}$ and $Rs_{3+4}$ very long (vs. much shorter in other genera of Orthophlebiidae) $Rs_1$ ascending, $Rs_{1+2}$ and $Rs_2$ almost at the same level. Forewing with Sc long and simple, reaching pterostigmal area; M with five branches; two crossveins present between $M_4$ and $Cu_1$; two anal veins. Hind wing with Sc single and simple, only reaching $M_{3+4}$ forking; M with four branches; one anal vein. Abdominal segments VI–VIII moderately elongate; very short notal process on tergite III, not extending beyond end of this tergite; postnotal process absent; two hirsute postnotal areas on tergite IV; tergites VI and VII with a pair of distal appendages each. Genital bulb without stem part, gonostyli long and thin.

*Burmorthophlebia multiprocessa* Soszyńska-Maj, Krzemiński and Wang sp. nov.
(Figure 2A–H)
*Zoobank registration:* urn:lsid:zoobank.org:act:D4B00956-1785-43C2-8183-3372557BDE6B
**Etymology**. Species name highlights the unusual morphology of the holotype, possessing multiple processes on abdominal segments.
**Material**. Holotype MP/3721, male.
**Locality and age.** Kachin amber, Myanmar; Late Cretaceous: earliest Cenomanian (98.8 ± 0.62 Ma).
**Diagnosis**. As for the genus (*vide supra*).
**Description**. Body length 9.5 mm from frons to the end of gonostyli (**Figure 2A**). *Head* (**Figure 2G**). Three occipital bristles; rostrum 2 mm long, first five flagellomeres only slightly longer than wide, gradually becoming more elongate, distal visible flagellomeres 3× longer than wide; maxillary palp five segmented, palpomeres slim, ca. 4× longer than wide. *Thorax*. Legs with setae arranged in whorls, each bearing a pair of long tibial spurs; fifth tarsomere ca. 4× longer than fourth one, two pretarsal claws; first tarsomeres bear no swellings on any leg. *Wings*. Proximally narrow and significantly broadening distally, 9.5 mm long and ca. 3.5 mm wide; wing membrane bearing a round, dark pigmented spot in medial sector. *Forewing* (**Figure 2C**). Sc and $R_1$ reaching pterostigmal area, two crossveins between C and Sc; $Rs_{1+2}$ almost 3× longer than $Rs_1$, $Rs_{3+4}$ longer than $Rs_3$ and $Rs_4$; M with five branches reaching wing margin, Mb forking beyond the Rs fork, $M_{1+2}$ 9× longer than $M_{3+4}$, $M_{4a+b}$ 3.5× longer than $M_{3+4}$; two m4-cu1 crossveins, the most proximal slightly bending $M_4$; two crossveins between $A_1$ and $A_2$, $A_3$ not visible. *Hind wing* (**Figure 2D**) same as in forewing; Mb fused with $Cu_1$; second anal vein reduced or invisible. *Abdomen* (**Figure 2E and F**). Elongate, notal organ on abdominal tergite III wide at base and curved toward distal part; postnotal organ on tergite IV composed of two areas of dense

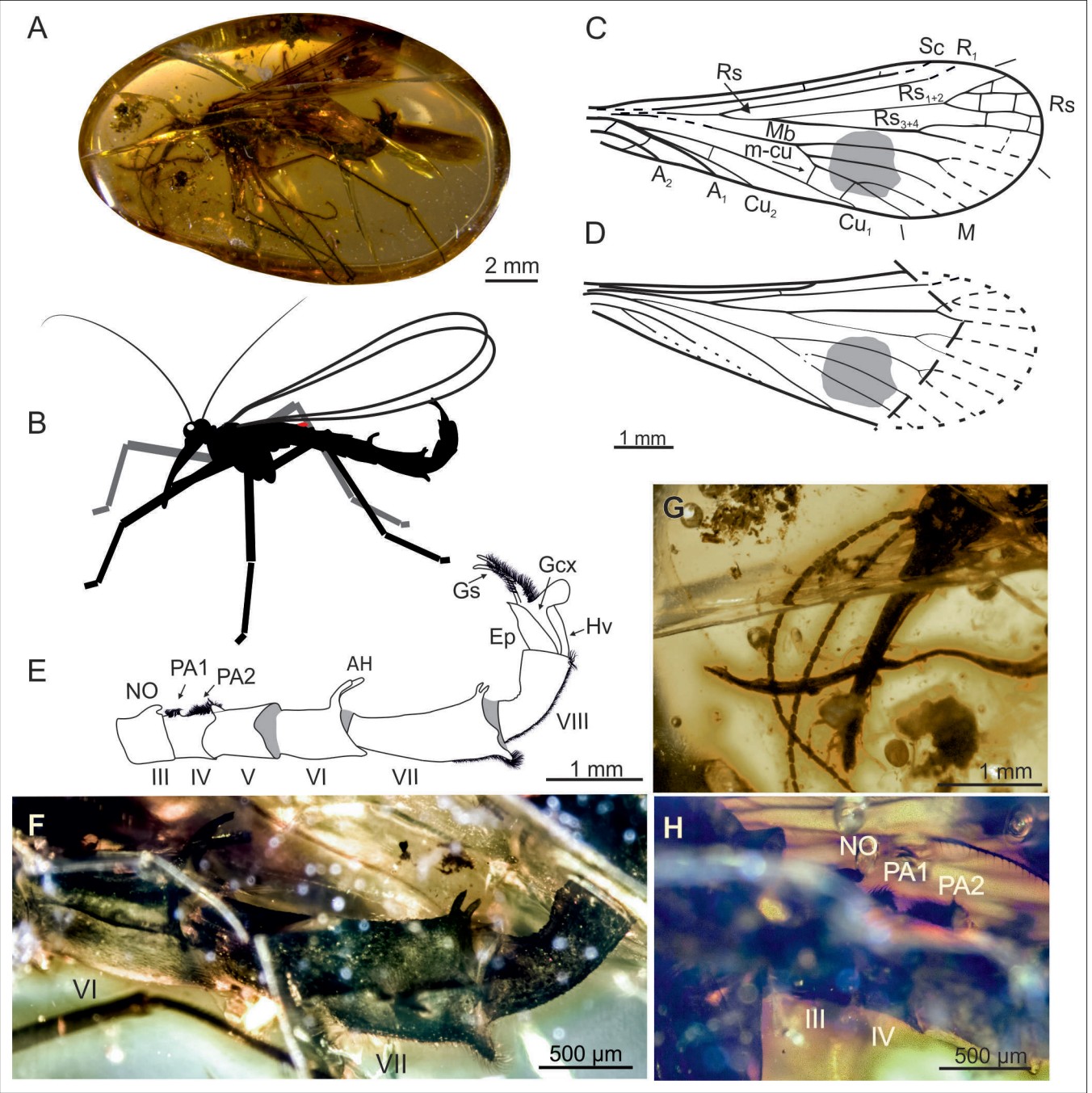

**Figure 2.** *Burmorthophlebia multiprocessa* gen. et sp. nov. (Orthophlebiidae) from Kachin amber, Myanmar (Late Cretaceous), holotype MP/3721. **A**, amber piece with habitus in lateral view; **B**, life reconstruction; **C**, forewing; **D**, hind wing, partially reconstructed; **E-F**, abdomen in lateral view, explanatory drawing (**E**) and photograph (**F**); **G**, detail of head in lateral view, antennae and rostrum; **H**, notal process and postnotal areas (on tergites III and IV, respectively). Abbreviations: AH, anal horns; PA1, PA2, postnotal areas 1, 2; other symbols as in *Figure 1*.

brush of short setae (called here postnotal areas 1 and 2), distal part of tergite IV curved upward and covered with dense setation (*Figure 2H*); abdominal segment V ca. 1/3 longer than wide; segment VI ca. ½ longer (800 µm) than wide, bearing a pair of relatively large anal horns on posterior margin of tergum; distinct sinuous seam between tergites and sternites (*Figure 2E and F*); tergites and sternites VII–VIII fused; segment VII (1.3 mm) distinctly thinned at base and gradually broadening distally, horns on posterior tergum margin, opposite to them the sternite bears a process with dense setae; segment VIII ca. 2× longer than wide, sternite covered with setae. *Genitalia*. Genital bulb narrow (*Figure 2E*);

gonostyli longer than gonocoxites, very thin at apex, covered with numerous setae; hypovalve broad; epandrium most likely with two processes.

**Remarks**. *Burmorthophlebia* gen. nov. differs from all other genera of the family Orthophlebiidae in number of veins in radial sector, and in unusually long veins $Rs_{1+2}$ and $Rs_{3+4}$ which are only terminally forked, at level of end of Sc. Wings of *Burmorthophlebia* gen. nov. are more narrow than in other orthophlebids. Five veins in radial sector are present also in a recently established family Protorthophlebiidae (**Soszyńska-Maj et al., 2019**), but the rostrum in this family is short (only c. twice as long as its maximum width), and VI-VIII abdominal segments in male are shorter than wide, while *Burmorthophlebia* gen. nov. has a long rostrum and elongate abdominal segments, both characters typical to orthoplebids.

## Cantabridae Soszyńska-Maj, Pérez-de la Fuente, Krzemiński and Wang fam. nov.

*Zoobank registration:* urn:lsid:zoobank.org:act:225FC0C1-5827-4CA1-9F44-B27F9EFD5681

**Type genus** *Cantabra* gen. nov., by monotypy and present designation.

**Diagnosis**. Head with three pairs of long ocellar bristles, rostrum elongate; pronotum elongate, longer than head, bearing long bristles; Sc single, costal area narrow, R and M sectors with four branches in both wings, crossvein cu2-a2 present in hind wings; male abdomen much longer than wings, with elongate and narrow segments VI–VIII, tergites and sternites VII and VIII fused; male gonostylus without median tooth, with several minute denticles.

**Remarks**. The wing venation and general habitus of *Cantabra* gen. et. sp. nov. resemble those of the Panorpidae and Panorpodidae. The new fossil species differs from panorpids in the distinct body bristles, particularly those on head and thorax, an unusually elongate pronotum, clearly longer than the head (shorter in panorpids), and the lack of median tooth on the inner surface of male gonostyli, instead bearing several minute denticles in a row. From panorpodids, *Cantabra* gen. et. sp. nov. differs in greatly elongate rostrum, elongate abdomen and fused VII and VIII abdominal segments. The remarkably elongate and narrow male abdomen of the new taxon resembles extinct family Holcorpidae, but differs from the latter in five veins in medial sector in both wings and less elongate abdominal segments VI-VIII (**Archibald, 2013**).

## *Cantabra* Soszyńska-Maj, Pérez-de la Fuente, Krzemiński and Wang gen. nov.

*Zoobank registration:* urn:lsid:zoobank.org:act:EF60B085-D327-4D60-BD97-DEC45E16E008

**Type species**. *Cantabra soplao* sp. nov., by monotypy and present designation.

**Etymology**. After the Latin 'cantabra' ('cantabrum' in neutrum) meaning 'from, or pertaining to, Cantabria' – that is, the Autonomous Community of Spain where the El Soplao outcrop is located. Gender is feminine.

**Diagnosis**. As for the family (*vide supra*).

## *Cantabra soplao* Soszyńska-Maj, Pérez-de la Fuente, Krzemiński and Wang sp. nov.

(Figures 3A–G and Figures 4A–E)

*Zoobank registration:* urn:lsid:zoobank.org:act:7DB2D8E1-3E2B-427A-8E3E-6BBFE3E6413F

**Etymology**. After the El Soplao amber outcrop, where the amber inclusion was found. The specific epithet is treated as a noun in apposition.

**Material**. Holotype CES-437, male.

**Locality and age**. El Soplao amber outcrop; middle Albian, Early Cretaceous (ca. 105 Ma).

**Diagnosis**. As for the genus (*vide supra*).

**Description**. Body length ~9.25 mm without antennae (*Figure 3A*). Head (*Figure 3B and C*). Rostrum with preserved proximal part about 1.5× as long as that of head; 20 flagellomeres preserved, all elongate, 2–3× as long as wide, covered with microsetae; three pairs ocellar bristles on vertex, ~ 350 μm long, probably surrounding ocelli (the latter not discernible). *Thorax*. Pronotum with long bristles, ~ 300–350 μm long; only one right foreleg partially preserved, elongate, rather delicate, bearing microsetae on its entire length; femur bearing ventral bristles; tibiae with a few whorls of spike-like setae arranged in whorls and bearing two long spurs distally. *Wings*. Only right wings fully preserved.

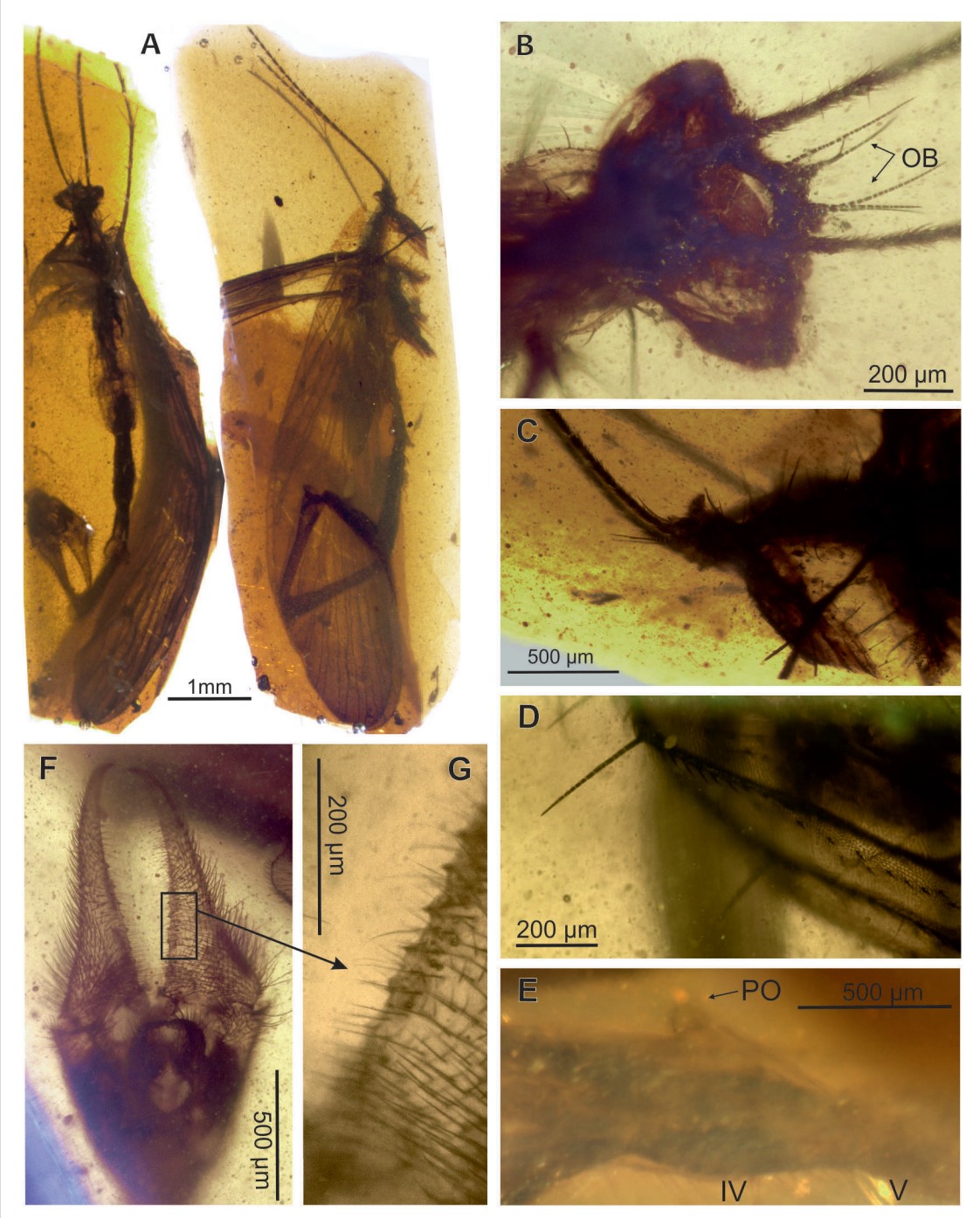

**Figure 3.** Photomicrographs of *Cantabra soplao* gen. et sp. nov. (Cantabridae fam. nov.), from El Soplao amber, Spain (Early Cretaceous), holotype CES-437. **A**, ventral and lateral habitus, respectively; **B-C**, head, frontoventral view (**B**), lateral view (**C**); **D**, setae on anal area of forewing; **E**, postnotal organ seen in ventrolateral view (head is to the left), morphology distorted due to the skewed view; **F**, genital bulb in ventral view; **G**, detail of gonostylus, showing its inner denticles. Abbreviations: OB, ocellar bristles; PO, postnotal organ.

Wings elongate and narrow, almost 4× longer than wide, 6.5 mm long, 1.7 mm wide; C covered with setae, longer and thinner in anal region. *Forewing* (*Figure 4A*). Costal area narrow, with Sc unbranched and long, reaching wing margin opposite tip of $M_4$; $R_1$ faint, its two branches surrounding pterostigmal area; radial sector with four branches; $Rs_{1+2}$ almost twice as long as $Rs_{3+4}$; $M_{1+2}$ almost 7× longer than $M_{3+4}$; $Cu_1$ and $Cu_2$ fused at base; three anal veins present, $A_1$ reaching the wing margin

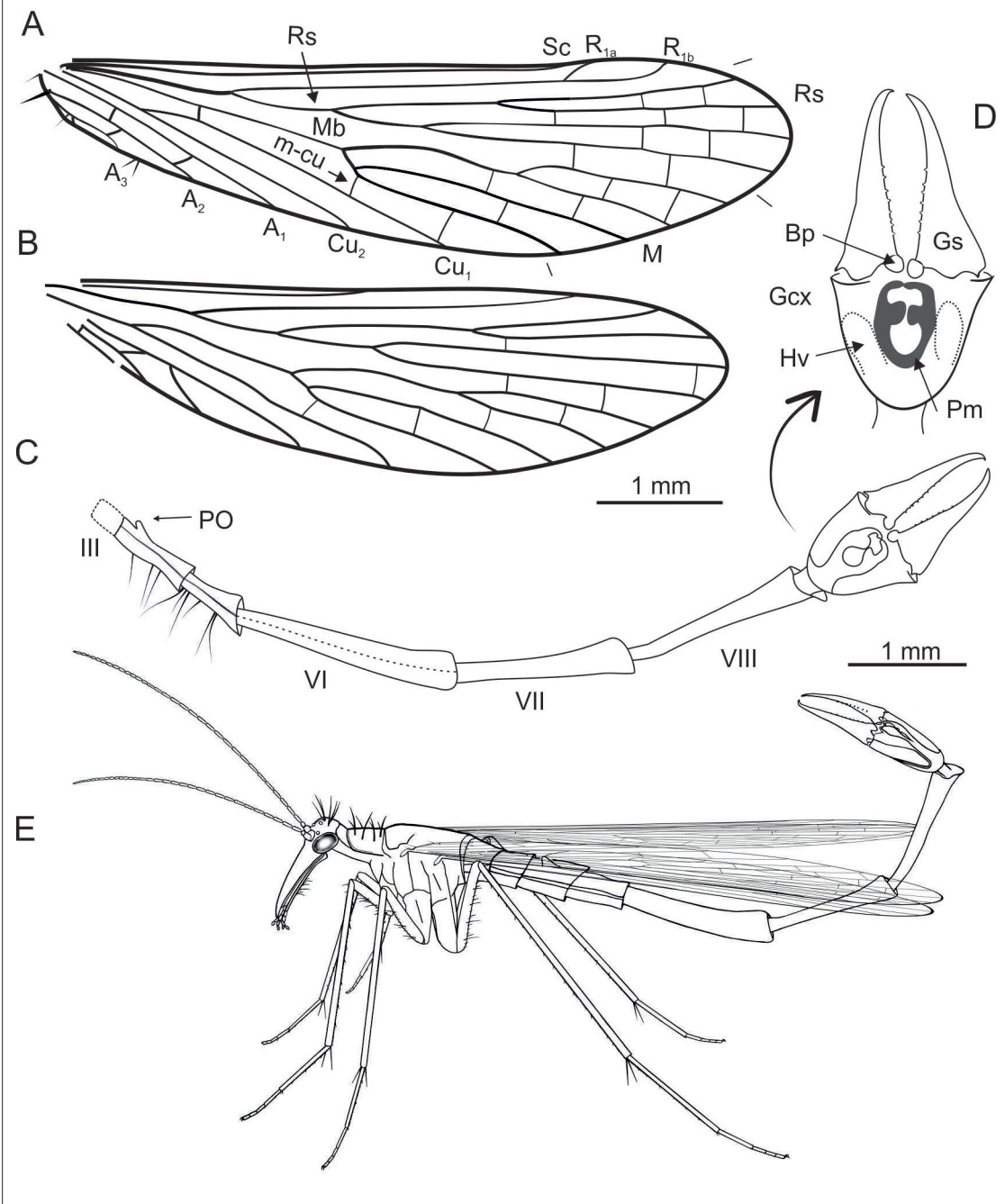

**Figure 4.** Explanatory drawings of *Cantabra soplao* gen. et sp. nov. (Cantabridae fam. nov.) from El Soplao amber, Spain (Early Cretaceous), holotype CES-437. **A**, right forewing; **B**, right hind wing (same scale for both wings); **C-D**, reconstruction of abdomen in lateral view, with genital bulb shown in ventral view (**C**); inset (arrow) shows the genital bulb expanded, with elements tagged (**D**); **E**, life reconstruction, in resting position. Notal organ not visible in holotype and thus not depicted for caution. Length of antennae and rostrum are conjectural; morphology of legs is based on the preserved left foreleg, which is mostly complete. Abbreviations: Bp, basal process; Pm, parameres; other symbols as in *Figure 1*.

beyond fork of Rs; short spines along anal veins and wing membrane on anal area; jugal area with few long bristles. *Hind wing* (**Figure 4B**). Venation mostly similar to that of forewing, except short Sc, not reaching beyond the level of end of $Cu_1$; $R_1$ forking at the level of end of $Cu_1$; Mb fused with $Cu_1$; $A_1$ proximally fused with $Cu_2$ and reaching wing margin only slightly beyond fork of Rs; one crossvein between $Cu_2$ and $A_2$; crossveins hardly visible. *Abdomen* (**Figure 4C**). Significantly longer than wings; tergite III not visible dorsally, notal organ not discernible, but (if present) not long, (otherwise it would

be visible above tergite IV); small postnotal organ present on tergite IV; long bristles on sternites IV–V; abdominal segments VI to VIII elongate and narrow, segment VI (1.8 mm long) 7× longer than wide, segment VII (1.4 mm) 6× longer than wide, segment VIII (1.5 mm) 10× longer than wide; segment VI most probably with a fusing seam; segments VII and VIII with tergites and sternites fused. *Genitalia* (*Figures 3F, G and 4C, D*). Genital bulb elongate, hypovalves probably broad, not extending to apex of gonocoxites; gonostyli (1 mm) longer than gonocoxites (700 μm), slender, inwardly curved at apex, without median tooth, and with rounded and relatively small basal processes; inner margin of gonostylus with at least eight short triangular denticles in a row along median and basal parts; basal processes of parameres long and curved ventrally, with their basal branch rounded, covered with dense microsetae reaching apex of gonocoxite.

## Discussion

A series of clamping devices used to seize and hold different body parts of the female during courtship and copulation exist in panorpid males. These structures ensure the eventual genital connection and sperm transfer and extend copulation time (*Zhong and Hua, 2013a*; *Zhong et al., 2014*). The latter is critical for males, as the more prolonged the copulation, the more sperm can be transferred and, thus, the greater the number of potentially fertilised eggs (*Engqvist and Sauer, 2002*; *Kullmann and Sauer, 2005*). Aside from genital clamping structures (among which the most conspicuous is a pair of pincer-like gonostyli), panorpids present non-genital clamping systems on their abdomens. These are notal and postnotal organs, as well as anal horns which are used to catch and immobilize female during mating. Based on the size, shape and interlocking mechanism of these clamping structures, the grasp on the body of female becomes more or less secure, therefore facilitating a greater degree of coercion during mating, which has been associated with less investment in costly nuptial gifts (*Tong and Hua, 2019*).

### Notal and postnotal organs

The notal organ (NO) is a process on abdominal tergite III. The postnotal organ (PO) typically consists of a small process on abdominal tergite IV (*Crampton, 1931*; *Mickoleit, 1971*). In extant scorpionflies, both structures are present in most Panorpidae, the only recent species from the Eomeropidae (the 'living fossil' *Notiothauma reedi* MacLachlan, 1877 [*MacLachlan, 1877*; *Crampton, 1931*]), and can be vestigial, if present, in the Panorpodidae (*Tong et al., 2017*; *Wang and Hua, 2018a*). Both NOs and POs act jointly by typically creating a clamping structure that grasps the female forewing before and during copulation; in panorpids, these organs allow to achieve and maintain the typical V-shaped mating position (Figure 6A and B; *Byers and Thornhill, 1983*). Moreover, the NO and the PO bear specialised setae, cephalically directed and thinned terminally, which interlace and/or electrostatically interact with both venational setae and membrane microtrichia from the female's wing (*Zhong and Hua, 2013a*). Observations on three species of *Neopanorpa* van der Weele, 1909 (*van der Weele, 1909*) bearing NOs of different lengths have shown different mating strategies in these species (*Zhong and Hua, 2013a*; *Tong and Hua, 2019*). Males of two species with short and medium-sized NO, i.e. *N. lui* Chou and Ran, 1981 (*Chou et al., 1981*) and *N. carpenteri* (*Cheng, 1957*) indirectly transfer a nuptial gift of a salivary secretion prior to copulation. The short NO of *N. lui* ends at the base of the PO, and the medium sized NO in *N. carpenteri* reaches the end of the abdominal segment IV; both stabilize the female wings only partially and apparently play an auxiliary role during copulation. Contrarily, *N. longiprocessa* (*Hua and Chou, 1997*), with a long NO capable to cover the entire wing width of the female, exclusively relies on coercive copulation, and does not offer nuptial gifts. Short NOs and POs pinch only the anterior margin of female forewings, and both processes are spatially suited to fit the latter. In *Panorpa amurensis* MacLachlan, 1872 (*MacLachlan, 1872*) (Figure 5A,B), the space created by the clasped NO and PO matches the space of the anterior valley fold of the female wing delimited by the convex costal vein (C), the concave subcostal vein (Sc), and the convex radial vein ($R_1$), clamping together with the PO before or at the latter vein. Our measurements of these distances in this and other panorpid species with various NOs indicate that this seems to be an effective arrangement to grip the anterior wing fold of the female (see *Supplementary file 1* for measurements). As the C, Sc, and $R_1$ run almost parallel over a long stretch, the female forewing may be gripped more anteriorly or posteriorly, usually by its half or basal third, as

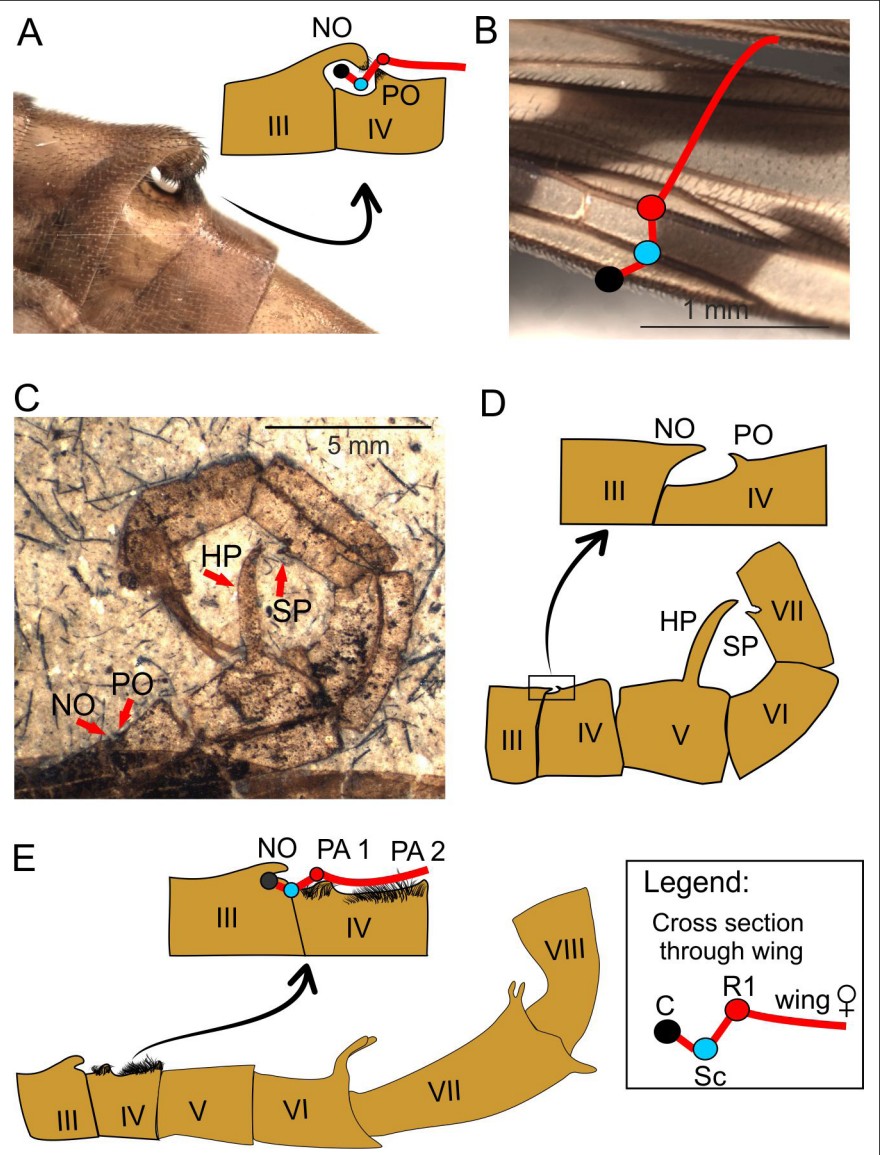

**Figure 5.** Spatial coupling between short male notal organs and female (fore)wings during mating in extant and extinct Mecoptera. **A–B**, *Panorpa amurensis* MacLachlan, 1872 (Recent): **A**, photograph of notal organ in male, and diagram cross-section through the notal and postnotal processes with anterior fold of female forewing entrapped; **B**, photograph and diagram of anterior fold in female forewing; **C-D**, *Orthophlebia heidemariae* Willmann and Novokschonov, 1998 (Late Jurassic): habitus (**C**) and explanatory drawing (**D**, inset showing notal-postnotal system magnified); **E**, *Burmorthophlebia multiprocessa*, gen. et sp. nov (Late Cretaceous): diagram of abdominal modifications (bottom) and hypothetical spatial coupling between male notal organ and female wing (top). Abbreviations: SP, small process (tergite VII); other as in *Figures 1 and 2*. Colour legend of cross section through veins: black circle, C (Costa); blue circle, Sc (Subcosta); red circle, R1 (1st radial vein); red line, wing membrane.

can be observed in photographs of mating panorpids (*Zhong and Hua, 2013a*; *Zhong et al., 2014*; *Tong and Hua, 2019*).

A short NO of the type similar to that in Recent panorpids was already present among the Late Jurassic panorpoids, as evidenced by *Orthophlebia heidemariae* (*Willmann and Novokschonov, 1998*; *Figure 5C, D*). This species is currently classified in Orthophlebiidae, a Mesozoic polyphyletic panorpoid assemblage ancestral to Panorpidae (*Soszyńska-Maj et al., 2019*). In *Burmorthophlebia multiprocessa* gen. et sp. nov. the NO, although also short, differs from all the hitherto known in Mecoptera, extinct or extant. The NO is distinctly raised over abdomen but poorly pronounced, not reaching any process to form a clamp with. The PO is here replaced by two elevated areas on tergite

IV, the postnotal areas 1 and 2 (PA1 and PA2; *Figure 5E*). The former is mound-like, located after an anterior depression on tergite IV and at a distance from the NO comparable to that of C to $R_1$. The PA2 is sharply elevated, and measurements show that it contacted the female wing just at or slightly beyond the concave radial sector vein (Rs). Both PAs were involved in adhering the female's wing as indicated by their location and specialised setation, consisting of short and dense tufts of hairs different from other abdominal bristles. Apparently, only the costal vein and a brief anterior wing membrane portion were covered by the NO in *B. multiprocessa* gen. et sp. nov., and so the remaining portion of the female wing was to a certain degree immobilised by the setation of both PAs. Among the extant panorpids the hirsute PAs, more flattened than those of *Burmorthophlebia* gen. et sp. nov. and situated on three consecutive tergites, have been described in *Neopanorpa setigera* Wang and Hua, 2018 (*Wang and Hua, 2018b*) and *N. luojishana* Wang and Hua, 2019 (*Wang and Hua, 2019b*), and also exist in *N. longistipitata* Wang and Hua, 2018 (*Wang and Hua, 2018b*). However, in these species the NO is long and able to keep the entire wing of a female pressed against the male's abdomen to a lesser or greater extent, so the hirsute PAs are probably only of auxiliary importance (*Figure 6A–D*). This contrasts with the armature in *B. multiprocessa* gen. et sp. nov., where it was the interaction between NO-PO setae and the integumentary structures of the female's wing that kept a larger portion of the latter in place, although the fixing must have been weaker. Recent panorpids with a short NO (and anal horns) adopt a mixed mating strategy (*Neff and Svensson, 2013*). After a courtship ritual, males present nuptial gifts to the female, but if they are not able to get one ready (including if they have been stolen by other males) or the gift is deemed as unsatisfactory by the female (e.g. small salivary secretions due to poor condition of the male), they resort to attempt coercive copulation (*Thornhill and Sauer, 1991*). *Burmorthophlebia multiprocessa* gen. et sp. nov., with its short NO and no other armature to immobilize the female's wing, probably had a similar approach when mating.

As introduced above, long NOs differ mechanically from short NOs. They are present in a small group of species classified in the genera *Panorpa* and *Neopanorpa* (*Hu et al., 2015*; *Tong and Hua, 2019*; *Wang and Hua, 2018a*). Instead of a pincer-like clamp composed by a PO and NO, the long NO covers the entire female's wing across its full width and immobilizes it more or less pressed against the abdomen. As the mating couple of *N. longistipitata* shows (*Figure 6A–D*), the leading edge of the female wing is also accommodated in a small depression of tergite IV under the base of the NO. As can be deduced from measured distances, of the three hirsute PAs on tergites IV–VI the first one supports the wing fold by the convex $R_1$; the more distal part of the wing is pressed against the abdomen by the long NO with the help of special muscles (*Thornhill, 1990*). The tip of the NO is locked between the genital hypovalves. Although this conformation fully encloses the female wing during mating, the resulting grip is likely not particularly tight. In any case, nuptial gifts are not offered prior or during the mating of *Neopanorpa* species with a long NO, such as *N. longiprocessa* (*Zhong and Hua, 2013a*), which would suggest that such process enables highly coercive mating behaviour.

Among extinct mecopterans, an extremely long NO was hitherto only known in *Baltipanorpa damzeni* from Baltic amber (*Krzemiński and Soszyńska-Maj, 2012*). In this species, the NO forms together with an unusually elongate PO a clip-like structure able to reach beyond the full width of the female's wing during mating (*Figure 6E*). Both NO and PO possess short tooth-like bristles, which would have increased their holding power. The PO is bent anteriorly, partially securing the clip in this way. *Baltipanorpa oppressiva* sp. nov has a similarly characteristic NO-PO clip bearing inner tooth-like bristles, but its NO is terminally forked in two prongs bent posteriorly at an almost right angle, which embrace the tip of PO (*Figures 1B, E and 6F*). The interaction of these structures in the new species results in a full clasp-like mechanism. The female wing enclosed in this remarkable clamp was likely fully immobilised, a withdrawal attempt probably causing risk of tearing the wing membrane. A similar function, although less effective than that in *B. oppressiva* sp. nov, must have been accomplished in *B. damzeni* by a few pairs of elongate bristles on the posterior side of the NO (*Figure 6E*). Furthermore, the NO-PO clamp in *Baltipanorpa* is raised at angle of about 50° from the abdomen, and so the captured female's forewing was also kept raised and separated from the male's abdomen. This position is deemed as genuine (not preservational) as it is now known from two different species (*Figure 6E and F*) and is unique to this genus. In extant panorpids with very long NOs (*Neopanorpa*), this process is placed at a much lower angle and keeps the female wing rather flat against the abdomen, having

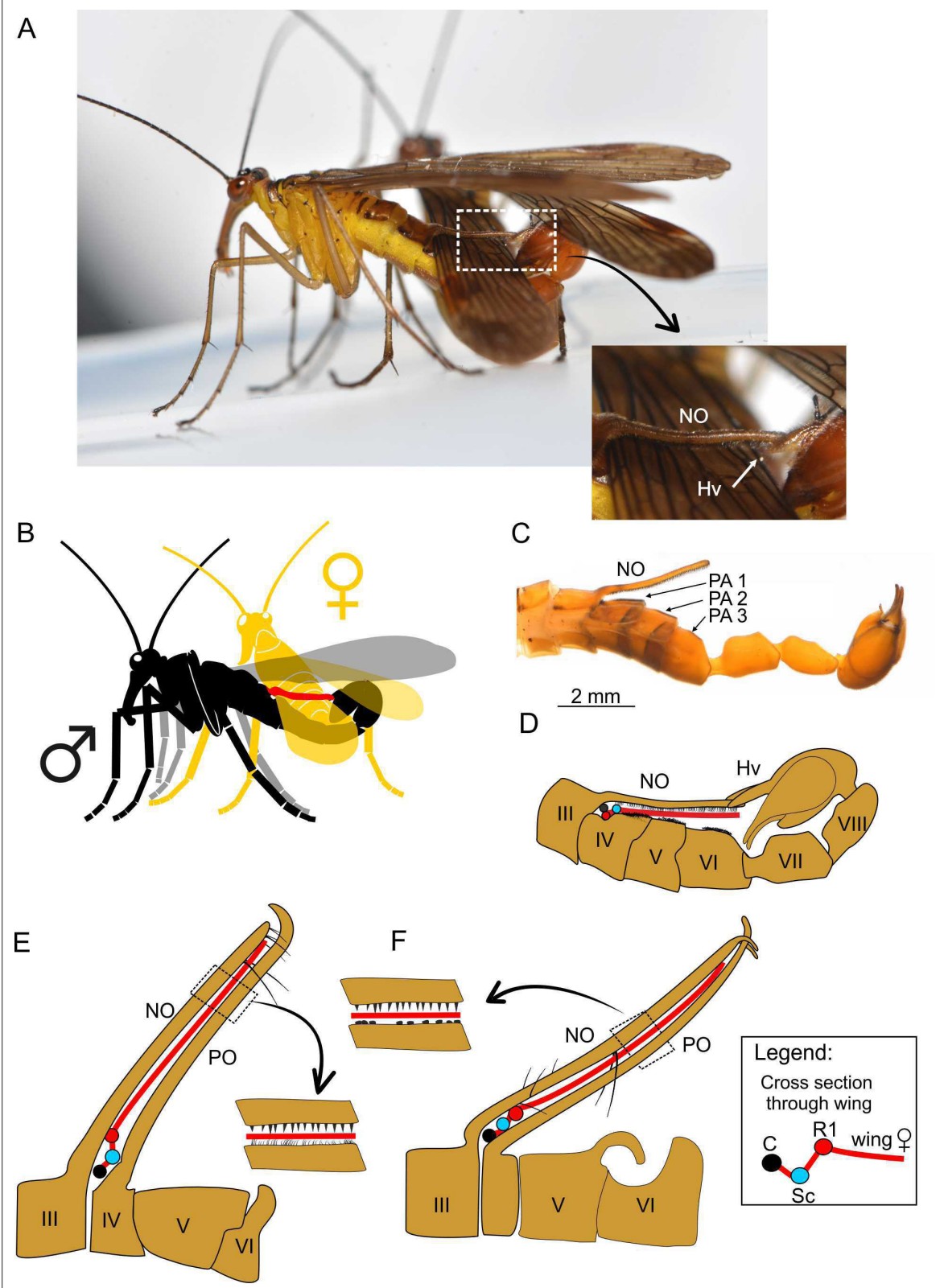

**Figure 6.** Spatial coupling between long male notal organs and female (fore)wings during mating in extant and extinct Mecoptera. A–D, *Neopanorpa longistipitata* Wang and Hua, 2018 (Recent): **A**, photograph of mating couple; inset (arrow) shows notal and hypovalves connection magnified; **B**, diagram of mating couple (notal organ depicted in red); **C**, photograph of male abdomen in lateral view; **D**, diagram of coupling mechanism between male notal organ and genital bulb; **E, F**, spatial coupling diagram between male notal-postnotal clip clasping the female wing in two species of *Baltipanorpa* (Eocene): *B. damzeni* Krzemiński and Soszyńska-Maj, 2012 (**E**); *B. oppressiva* sp. nov. (**F**). Abbreviations as in *Figures 1, 2, 4 and 5*.

no other counterpart such as the long PO in the two fossil *Baltipanorpa* species discussed herein (*Figure 6A and D*).

## Anal horns

The anal horns are distal processes directed posteriorly on abdominal tergite VI. Among extant panorpids, anal horns are single (in *Cerapanorpa* Gao, Ma and Hua, 2016 (in *Gao and Hua, 2016*), *Megapanorpa* Wang and Hua, 2018 (*Wang and Hua, 2018a*), and some species of *Panorpa* Linnaeus, 1758), or paired *Dicerapanorpa* Zhong and Hua 2013 (*Zhong and Hua, 2013b*), and arise at different angles from the dorsal end of tergite VI (*Zhong et al., 2014*; *Tong et al., 2018*; *Wang and Hua, 2019a*). Aside from panorpids, anal horns are also known in *Notiothauma reedi* (Eomeropidae; *Crampton, 1931*; *Mickoleit, 1971*). In conjunction with tergite VII, which is proximally narrowed and/or depressed, the anal horns create a clamping system to accommodate and hold the female's terminal abdomen into the strained copulatory position necessary for sperm transmission. It has been demonstrated that anal horns play an important role at initiating copulation and in prolonging its duration in the absence of nuptial gifts and under female resistance, and that the lack of anal horns is correlated with the inability to prolong copulation after gift provisioning (*Zhong et al., 2014*).

*Burmorthophlebia multiprocessa* gen et sp. nov. bears a pair of anal horns raised at a 45° angle (*Figures 2E, F and 5E*). By comparison with extant species, these processes must have been involved in manipulating and holding the female abdomen during mating. Paired anal horns in other fossil species are known in *Orthophlebia longicauda* Willmann and Novokschonov, 1998 (Orthophlebiidae; Late Jurassic of Karatau in Kazakhstan) and in two species of *Holcorpa* Scudder, 1878 (Holcorpidae) from two Eocene localities (Florissant, Colorado, USA, and the Okanagan Highlands, British Columbia, Canada) (*Scudder, 1878*, *Willmann and Novokschonov, 1998*; *Archibald, 2013*).

## Other mating-related modifications in males

Aside from notal-postnotal organs and anal horns, other male abdominal processes involved in the physical interaction with the female during mating are occasionally present in scorpionflies. Among extant panorpids these are positioned on the ventral side, such as a long process on abdominal sternite III (*Neopanorpa furcata* (Hardwicke, 1823)) and a 'ventral hook' on sternite VI (*Leptopanorpa linyejiei* Wang and Hua, 2020) (*Hardwicke, 1823*; *Wang and Hua, 2020*). In fossil panorpoids, other male abdominal structures with assumed clamping function occurred in the Late Jurassic species *Orthophlebia heidemariae*. This was the case of a large medial horn-like process on tergite V (*Figure 5C and D*). Such process has been referred to as 'tergal spine', 'unpaired median tergal process (monocornus)', or a 'median tergal horn' (*Crampton, 1931*; *Mickoleit, 1971*; *Willmann and Novokschonov, 1998*, respectively). It is probable that this process was also used to fix the female abdomen during mating in conjunction with other more distal structures from the male dorsal abdomen. The abdomen of *O. heidemariae* is preserved in an upturned position and forming a full circle (*Figure 5C*), which is a frequent mating position among panorpid males. The large horn-like process is opposite to a small putative process and notch on tergite VII (*Figure 5D*); both structures likely interlocked with one another to facilitate a strained copulatory position (*Willmann and Novokschonov, 1998*). In *Baltipanorpa oppressiva* sp. nov. the horn-like process on tergite V is strongly recurved, and the subsequent tergite VI has an anterior depression and a greatly expanded end, resulting in a notched shape. Both specialised shapes strongly suggest that this horn and the tergite VI were engaged in holding the female abdomen during mating (*Figure 6F*). Such conformation was enabled to a certain extent by the position of the female's wings (at least a forewing) which was raised high (56°) over the male's abdomen and secured in this position by the extremely long notal-postnotal clip unique to *Baltipanorpa*. Moreover, *Burmorthophlebia multiprocessa* gen. et sp. nov. bears paired processes directed posteriorly at 45° angle on tergite VII, similar to the paired anal horns present on tergite VI, but smaller. Alleged homologous structures have been formerly referred to as 'paired tergal processes (bicornua)' or 'lateral tergal horns' (*Crampton, 1931*; *Mickoleit, 1971*). The most plausible function for these processes in this new species was also holding the female terminal abdomen during mating.

*Cantabra soplao* gen et sp. nov. has greatly elongate abdominal segments VI–VIII. Among extant panorpids, such development is present in species of *Leptopanorpa* MacLachlan, 1875 (*MacLachlan, 1875*) endemic to Indonesia, and also in a few *Neopanorpa* species (*Wang and Hua, 2020*). Although biology of these taxa remains unknown, the greatly elongate abdomens are considered to be

influenced by sexual selection, in displays to females and competition among male rivals. Extremely elongate abdomens were also present among other extinct panorpoid lineages: the Orthophlebiidae (*Willmann and Novokschonov, 1998*), the Holcorpidae (*Archibald, 2013*; *Li et al., 2017*; *Zhang et al., 2021*), and in taxa of uncertain relationships from the Middle Jurassic of China, where abdomen elongation was extreme (*Wang et al., 2013*). Great abdominal elongation is regarded as a result of convergence (*Wang and Hua, 2020*). Additionally, in the Holcorpidae and Orthophlebiidae swellings on the first tarsomeres of the hind legs in males were most probably used in sexual display; the size and shape of these swellings was likely species specific (*Zhang et al., 2021*). In the mecopterans presented herein the tarsomeres were not modified (yet in *Cantabra soplao* gen. et sp. nov. the shape of legs is unknown). Tarsal swellings do not occur in Recent Panorpoidea.

## Conclusions

The morphology of the male's abdomen in extant scorpionflies represents a good approximation to the mating strategy used toward a female. Thus, the abdominal structures of fossil specimens have the ability to convey information on the likely reproductive strategies of extinct scorpionflies and, to some extent, on their evolution. Based on the disparate male abdominal shapes of the three fossil taxa described herein, these almost certainly had different mating strategies along a nuptial gifting-coercive gradient as observed among their extant panorpid relatives. Firstly, the Cretaceous species *Cantabra soplao* gen. et sp. nov., currently the only known member of its lineage, bears a small PO, with the absence of other posterior abdominal processes. This suggests that its mating behaviour was devoid of coercion and that nuptial gifting must have played an important role in these scorpionflies. Extant panorpid species which NO-PO system is very small or even absent and lack posterior abdominal processes ('horns') either rely on nuptial edible gifts or on mouth-to-mouth feeding during copulation adopting an O-shaped position instead of the typical V-position (*Figure 6A*; *Zhong et al., 2015*). It is plausible that the mating in *C. soplao* gen. et sp. nov. was similar. Regardless, the great degree of elongation of the abdomen in this species was probably involved in sexual selection dynamics.

Secondly, the Cretaceous *Burmorthophlebia multiprocessa* gen. et sp. nov., the only orthophlebiid specimen described from amber to date, likely exhibited a mixed mating strategy involving both nuptial gifting and coercive behaviour, as observed in some extant Panorpidae species. This inference is based on a weak notal-postnotal clamping system and two sets of paired processes on the abdomen, which likely ensured female restraint and probably extended copulation beyond any possible nuptial gift stages. Noteworthy, abdominal processes are exceptionally abundant in this species, on the dorsal and ventral sides. Such armature is unknown among extant Panorpidae, and can be compared only with that of *Nothiotrauma reedi* of the relict family Eomeropidae.

Lastly, in the Eocene *Baltipanorpa oppressiva* sp. nov. the NO-PO wing clamp, which resembled a sealing clip, was able to tightly grasp the full wing width of the female. Unlike its previously known congeneric relative *B. damzeni*, this structure bore a sophisticated terminal clasp. Together with a highly recurved abdominal horn in this new species, these traits represent the most extreme set of female clamping devices known among scorpionflies, both extinct or extant. Consequently, a fully coercive mating strategy was likely dominant, even exclusive, in this species. During copulation, the female wings (at least the forewings) were likely kept in the notal-postnotal clamp well raised over the male abdomen, this way being separated from the male's anal horn, which is positioned more proximally in this species (i.e. on tergite V vs. in tergite VI in other panorpids). This inclined position of the female's wing(s) enabled the entrapment of her abdomen by the male's anal horn during mating. In evolutionary terms, the fitness costs of such organs and the resulting coercive mating behaviour might have been too high for *Baltipanorpa*. The very long notal and postnotal processes (*Figure 6E and F*) raised over the male's body might have hindered flight ability; moreover, because of their size (and perhaps their permanently raised position) these processes were likely prone to injury and thus they might entail a risk for their bearers. These factors might have contributed to the extinction of *Baltipanorpa* and its singular morphology, so that the evolution of this extreme clamping device resulted in a blind alley. Moreover, the biology of the female could have also contributed to this extinction if the mating became too oppressive or dangerous, related to the risk of tearing the wing membrane with the teeth and hard bristles of the male's notal organ.

The remarkable diversity of abdominal configurations shown herein suggests that panorpoid mecopterans had accordingly disparate mating strategies and behaviours in the past. Data based

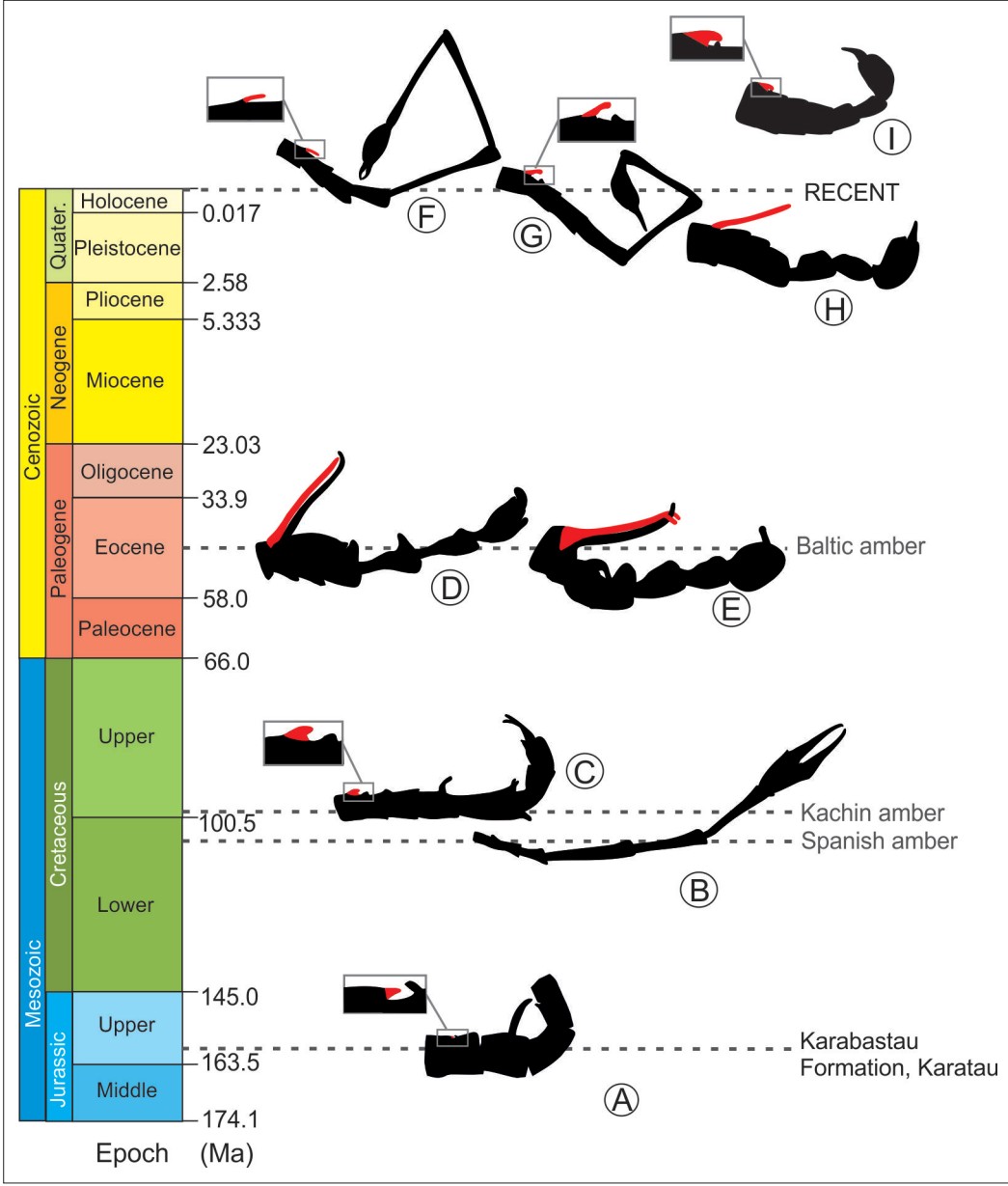

**Figure 7.** Male abdominal modifications in extinct and extant Panorpoidea on a time scale: **A**, *Orthophlebia heidemariae* Willmann and Novokschonov, 1998 (Orthophlebiidae); **B**, *Cantabra soplao* gen et sp. nov. (Cantabridae fam. nov.); **C**, *Burmorthophlebia multiprocessa* gen. et sp. nov. (Orthophlebiidae); **D**. *Baltipanorpa damzeni* Krzemiński and Soszyńska-Maj, 2012 (Panorpidae); **E**, *Baltipanorpa oppressiva* sp. nov. (Panorpidae); **F-I**, some examples of extant Panorpidae species: **F**, *Leptopanorpa linyejiei* Wang and Hua, 2018; **G**, *Leptopanorpa jacobsoni* van der Weele, 1909; **H**. *Neopanorpa longistipitata* Wang and Hua, 2018; **I**. *Panorpa jinhuaensis* Wang and Hua, 2019. Notal processes depicted in red. Drawings not at same scale.

on fossil and extant species suggest that the coercive mating behaviour in this group is secondary (derived) from the habit of nuptial gifting. According to the current fossil record, long notal organs did not evolve prior to the Eocene (*Figure 7*; *Supplementary file 2*). Data from extant mecopterans point at the Choristidae as the sister group of Panorpoidea (*Willmann, 1989*; *Wang and Hua, 2021*), which lack notal organs or other abdominal modifications (including elongate distal segments) and adopt an O-shaped mating position while the male transfers mouth-to mouth salivary secretion to the female (*Zhong et al., 2015*). Hence, it is assumed that the latter condition is likely plesiotypic for Panorpoidea.

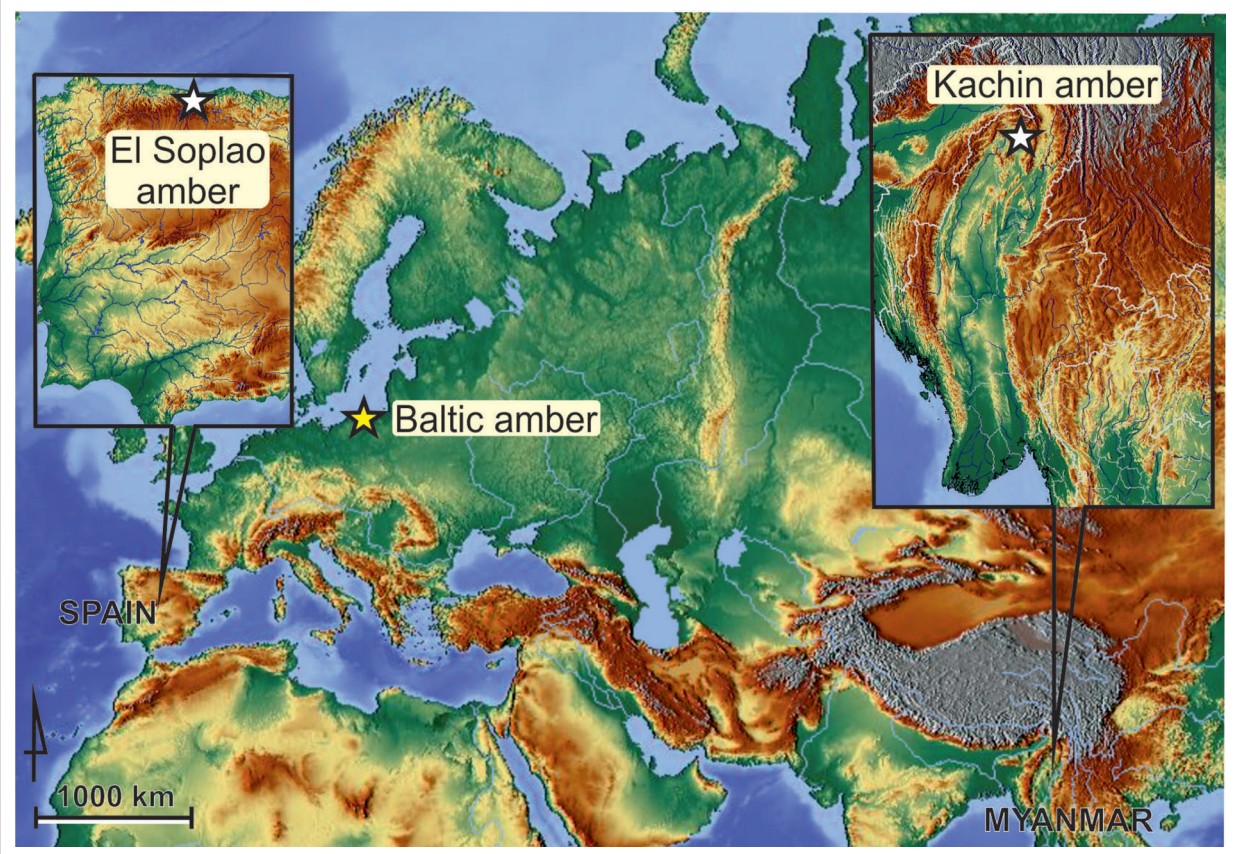

**Figure 8.** Geographical provenance of amber specimens studied herein. The two white stars represent Cretaceous localities (El Soplao amber, Spain; Kachin amber, Myanmar); the yellow star an Eocene deposit (Baltic amber).

Our data provide first steps toward gaining a deep-time perspective able to inform discussions on the evolution of mating-related sexual conflict in Mecoptera and in insects more generally. Extant fauna show that extreme female holding mechanisms such as those from *Baltipanorpa* did not survive until today among scorpionflies (*Wang, 2020*). In fact, mating strategies involving gift provisioning appear to have been evolutionarily favored among mecopterans contrary to forced mating, and likely more prone to promote the species' genetic diversity. In order to start formulating hypotheses about when mating strategies –including those fully coercive– and associated anatomical structures of scorpionflies appeared throughout evolutionary history, it will be necessary to keep discovering fossil males with good abdominal preservation. The latter will in turn necessitate from improving the knowledge on the currently contentious phylogenetic relationships among mecopteran groups.

## Materials and methods
### Geological context
The geographical provenance of the amber samples used in this study is marked on the map provided (*Figure 8*). The Spanish amber sample was gathered in official excavations with the participation of one of us (RPF) taking place between 2008 and 2010 in the El Soplao outcrop (near Rábago village, Cantabria, northern Spain) (*Najarro et al., 2009*; *Najarro et al., 2010*), and is dated as middle Albian, Early Cretaceous (ca. 105 Ma) (Barrón, pers. comm.). The Myanmar amber sample comes from Kachin, near Tanai town, Hukawng Valley, Kachin State, Myanmar, and is dated as earliest Cenomanian (98.8 ± 0.62 Ma), Late Cretaceous (*Shi et al., 2012*). It was acquired in 2016, prior to the armed conflict and the escalation of the ethnic strife in the area (*Haug et al., 2020*; *Szwedo et al., 2020*). The age of Baltic amber is estimated as Lutetian, middle Eocene (ca. 45 Ma) (*Grimaldi and Ross, 2017*), or as late Eocene (*Kasiński et al., 2020*).

## Specimen repository

All the studied specimens are deposited in public institutions. Holotype CES-437 is deposited at the Institutional Collection from the El Soplao Cave (Government of Cantabria), Celis, Cantabria, N Spain. The specimen is partly preserved, the amber piece was polished and included in Epoxy resin for visibility and protection. Holotype MP/3721 is housed at the collection from the Museum of the Institute of Systematics and Evolution of Animals, Polish Academy of Sciences, Kraków, Poland (ISEA PAS). The specimen is almost complete, several cracks cut the specimen. Holotype MP/2711 is housed at the collection of the Museum of the ISEA PAS, Kraków, Poland. The specimen is well preserved but incomplete, a neuropteran and a cecidomyiid dipteran are present as syninclusions.

## Methods

Photographs of amber inclusions were taken with a Leica M205C stereomicroscope and an attached Leica DFC295 camera, under a Nikon Eclipse E100 compound microscope with an attached Nikon DS F11 camera at the University of Łódź, and with a Nikon SMZ25 stereomicroscope equipped with a Nikon DS-Ri2 digital camera at the ISEA (PAS). In most instances, incident and transmitted light were used simultaneously. Stacks of photographs were processed using NIS-Elements Imaging Software. Drawings were obtained based on the photographs but corroborating the morphological details under the optic equipment, and then digitally processed in CorelDraw X10.

Adult extant scorpionflies were caught by one of us (JSW) with a collecting net and then preserved in 95% ethanol or pinned. These specimens are deposited at the Biological Science Museum, Dali University (DALU), Yunnan Province, China. Photographs of the insects were taken with a Nikon D7000 digital camera with a Nikkor AF-S Micro 105 mm f/2.8 lens. The figured specimens of *Panorpa amurensis* were collected by Sigitas Podenas in South Korea in 2014, and are deposited at the collection of the Department of the Invertebrate Zoology and Hydrobiology, University of Łódź, Poland.

Maps were built using the app Maps-For-Free (https://maps-for-free.com) and modified with the software programs Corel Draw and Corel Photopaint X7.

Wing venation terminology follows that of *Tillyard, 1933*, with minor modifications after *Soszyńska-Maj et al., 2018*. Wing vein abbreviations: A, anal vein; C, costal vein; Cu, cubital vein, M, medial vein; R, radial vein; R1, first radial vein; Rs, radial sector vein; Sc, subcostal vein. Other abbreviations used in figures appear in their respective captions.

This published work and the associated nomenclatural acts have been registered in ZooBank, the online registration system for the International Code of Zoological Nomenclature. The ZooBank LSIDs (LifeScience Identifiers) can be resolved and the associated information viewed through any standard web browser by appending the LSID to the prefix "http://zoobank.org/". The LSID for this publication is urn:lsid:zoobank.org:pub:391453EC-D289-41A6-9C62-C3A78617D012.

## Acknowledgements

We express our gratitude to the Editor and four Reviewers for the extensive comments. We thank the El Soplao Cave for providing access to the Spanish amber specimen. Drs Enrique Peñalver and Xavier Delclòs are thanked for managing the loan of that specimen and providing support and discussions, and Rafael López del Valle for its preparation. Sigitas Podenas is thanked for the gift of specimens of *Panorpa amuriensis*.

## Additional information

### Funding

| Funder | Grant reference number | Author |
| --- | --- | --- |
| National Science Centre, Poland | 2013/09/B/NZ8/03270 | Agnieszka Soszyńska-Maj Katarzyna Kopeć Wiesław Krzemiński |

| Funder | Grant reference number | Author |
| --- | --- | --- |
| National Science Centre, Poland | 2016/23/B/NZ8/00936 | Agnieszka Soszyńska-Maj<br>Ewa Krzemińska<br>Kornelia Skibińska<br>Katarzyna Kopeć<br>Wiesław Krzemiński |
| AEI/FEDER, UE | CGL2017-84419 | Ricardo Pérez-de la Fuente |
| Dali University | High-level Talents KY2096124040 | Ji-Shen Wang |
| National Science Centre, Poland | 2018/31/B/NZ8/02113 | Krzysztof Szpila |

The funders had no role in study design, data collection and interpretation, or the decision to submit the work for publication.

## Author contributions

Agnieszka Soszyńska-Maj, Conceptualization, Investigation, Methodology, Resources, Software, Validation, Visualization, Writing – original draft, Writing – review and editing; Ewa Krzemińska, Conceptualization, Validation, Visualization, Writing – original draft, Writing – review and editing; Ricardo Pérez-de la Fuente, Conceptualization, Investigation, Resources, Validation, Visualization, Writing – original draft, Writing – review and editing; Ji-Shen Wang, Investigation, Methodology, Resources, Validation, Visualization; Krzysztof Szpila, Conceptualization, Validation, Writing – original draft, Writing – review and editing; Kornelia Skibińska, Investigation, Software, Validation, Visualization; Katarzyna Kopeć, Data curation, Investigation, Project administration, Resources, Validation; Wiesław Krzemiński, Conceptualization, Funding acquisition, Investigation, Resources, Supervision, Validation, Writing – original draft, Writing – review and editing

## Author ORCIDs

Agnieszka Soszyńska-Maj http://orcid.org/0000-0002-2661-6685
Ewa Krzemińska http://orcid.org/0000-0002-3431-9963
Ricardo Pérez-de la Fuente http://orcid.org/0000-0002-2830-2639
Ji-Shen Wang http://orcid.org/0000-0002-0188-0228
Krzysztof Szpila http://orcid.org/0000-0002-3039-3146
Kornelia Skibińska http://orcid.org/0000-0002-5971-9373
Katarzyna Kopeć http://orcid.org/0000-0001-6449-3412
Wiesław Krzemiński http://orcid.org/0000-0001-5685-891X

## Decision letter and Author response

Decision letter https://doi.org/10.7554/eLife.70508.sa1
Author response https://doi.org/10.7554/eLife.70508.sa2

# Additional files

## Supplementary files

• Supplementary file 1. Spatial match between the space created by the clasped notal and postnotal processes (NO-PO) and the anterior valley fold (from Costa to R1, measured across wing) of the wing in some recent panorpids. T5, T6, tergites 5, 6. Measurements are approximate, based on scale bars in published figures as specified.

• Supplementary file 2. Remarks on the abdomen of fossil Panorpoidea. AH, anal horn; NO, notal organ; PA, postnotal area; PO, postnotal organ; A5, A6, abdominal segments 5, 6.

• Transparent reporting form

## Data availability

All data needed to evaluate the conclusions in the paper are present in the paper and/or the supplementary materials. Additional information related to this paper may be requested from the authors. Investigated fossils are available in public institutions: at the Institutional Collection from the El Soplao Cave (Government of Cantabria), Celis, Cantabria, N Spain and at the collection from the Museum

of the Institute of Systematics and Evolution of Animals (ISEA), Polish Academy of Sciences (PAS), Kraków, Poland.

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
