## [Editor Report]

The authors present, describe and interpret exquisite insect fossils preserved in amber. The detailed scorpionfly morphology reveals specific reproductive morphological adaptations that are clear enough to corroborate hypotheses explaining how these insects may have mated. Based on careful morphological comparisons with extant scorpionfly species, the authors are able to speculate on how sexual conflict evolved and shaped insect mating behavior. The findings are of interest to evolutionary biologists studying reproduction and behavior, entomologists and paleontologists. Given the unusual diversity in insect mating behavior, which is simultaneously fascinating and horrifying, this deep-time perspective is likely to interest not only colleagues in the field, but also the general public.

---

## [Decision Letter]

**Decision letter after peer review:**

Thank you for submitting your article "Evolution of sexual conflict in scorpionflies" for consideration by *eLife*. Your article has been reviewed by 4 peer reviewers, and the evaluation has been overseen by a Reviewing Editor and Molly Przeworski as the Senior Editor. The following individuals involved in review of your submission have agreed to reveal their identity: Ren Dong (Reviewer #1); Alexei Bashkuev (Reviewer #2); Dany Azar (Reviewer #3).

The reviewers have discussed their reviews with one another, and the Reviewing Editor has drafted this to help you prepare a revised submission. Please comply with the comments and suggestions to your best ability, mark all changes using a blue font to indicate changes in the revised manuscript, and respond in a point-by-point fashion. This is essential to enable the reviewing editor to fully evaluate the merit of your revision.

Essential revisions:

General points

The following revisions improve both the scientific presentation and clarity for *eLife*'s broad readership. Especially considering the paper has strengths in its unique fossil data, however, there is a need to more clearly and logically support the main points informed by the data. Several sections are unnecessary complex, which makes this manuscript unclear for both specialists and general *eLife* readers alike. Please address the issues thoughtfully and holistically keeping in mind *eLife* serves the entire biology community by publishing exceptionally clear and rigorously supported manuscripts. Any jargon deemed essential in the manuscript should be introduced in a more accessible fashion upon first use so a general reader can follow.

1) Not only the general readership needs improved higher level framing to enable a clear read. Fellow paleoentomologists would also greatly benefit from a paragraph in the introduction that includes other insects that follow similar strategies (e.g. Tettigoniidae, Megaloptera) together with some of the most relevant and general references. Considering the species found in Spanish amber belong to a new family, it will be challenging to make comparisons with recent relatives. Hence it may serve both the general and specialist reader to present the current findings in the context of mating strategies of other insects with sexual conflicts. To help introduce the general reader to the focus of this study, it would help to present an introductory figure showing this broader context, both in extant and extinct species, highlighting the unique contribution the new fossils make. Optimizing the graphic presentation and introducing key specialist terms in a clear fashion. To find examples for what a general reader would find clear consider how an in-depth National Geographic or New York Times Science article would communicate the context of the new finding and fossil as inspiration for designing an introductory figure. Given the high graphic design quality of the authors other figures, this may be relatively straightforward to add, it will greatly benefit all readers.

In this introductory figure and the associated introductory text, please consider there are numerous well preserved and diverse fossil scorpionflies (e.g. orthophlebiids sensu lato) from the Mesozoic Lagerstätten in China (Middle Jurassic Daohugou Biota, Early Cretaceous Jehol Biota) that are curated across institutions and museums. These collections have been studied in detail. Introducing this fact to the reader and presenting the new fossils in this established context is essential to clearly frame the new findings and show the scientific significance. In this regard the statement "in older fossil panorpoids, the abdomen is lacking or this portion is obscured" should be updated and clarified to avoid overstatement.

2) Regarding the relationship between gift-giving and coercive behavior. Most insects do not have nuptial gifts during the mating, coercive behavior is very common, please integrate this in the framing of Mecoptera as models of evolutionary research. The emergence of nuptial gifts seems to have greatly improved the diversity of reproductive behavior in scorpionflies. Considering this, coercive behavior may perhaps still be considered at the lower level of reproduction. In this light nuptial gifts, rather than coercion, may make scorpionflies an important reproductive evolutionary model. Given the discussion of this matter by the reviewers, it is helpful to clarify the manuscript and either integrate this perspective or provide a compelling rebuttal that the reviewers can evaluate to consider the authors preferred perspective.

3) The abstract could more clearly communicate the key findings and conclusions. Consider further highlighting the new evidence presented for the evolution of reproductive behavior in Mecoptera (/ the Holometabola). Please revise in a fashion that serves the general *eLife* reader.

4) The introduction should more carefully consider the number of extant species of scorpionflies, which is over 700. Consider that these scorpionflies providing saliva globes typically do not provide the dead arthropods at the same time. This has ramifications for the number of gifts that can be given at one time.

5) The descriptive and Discussion sections should be separated more clearly so the reader can digest the evidence before considering the authors evolutionary interpretation and hypothesis.

6) Present the new family, Cantabridae, in the context of other families of Panorpidea beyond Panorpidae. Consider comparing them further with the Holcorpidae based on their similarity in elongated abdomen. At a higher level, provide higher level context for the general *eLife* reader so they can follow the merit and relevance of considering the Cantabridae and contrasting them with other families through the lens of evolutionary diversity.

7) The discussion benefits from more clearly structuring the sections with informative subtitles that help the general *eLife* reader grasp the big picture points. Examples of dry and challenging concepts discussed include, Notal and postnotal organs; Anal horns; Other male abdominal processes; Male abdominal configurations and mating strategies.

8) Please better discuss why there may be both coercive behavior and nuptial gifting in some groups, what's the functional / evolutionary reason? Do more groups have a mixed strategy or only the orthophlebiid? Thoughtful explanation supported by key examples will help clarify this issue in the discussion.

9) Consider that females have several mating options and can choose to accept more than one male, informed by specific behaviors and morphology.

9) Given this study lacks a full-fledged phylogenetic analysis, consider deleting the discussion on phylogenetic relationships. Alternatively, consider providing a phylogenetic analysis of male abdomen morphology to strengthen the rigor of the study. Or if this is infeasible, but the authors wish to make a phylogenetic point, keep in mind that the only acceptable location for speculation is in the last paragraph of the discussion proposing future work in a few concise sentences (at most).

10) The mating behavior of Boreidae scorpionflies is well-described in literature, however, this context is not well-introduced and discussed. E.g. "notal and postnotal organs …can be vestigially present in Boreidae" is technically incorrect. The males of Boreidae have a non-homologous but functionally similar female-clamping device, which include hook-like rudimentary wings and structures on the 2nd and 3rd abdominal segments (vs on 4th and 5th in panorpids), usually referred to as tergal apophyses (e.g. Mickoleit and Mickoleit, 1976), that can be well-developed in some species. However, it is not apparent that this well-developed (better than in most panorpids) clamping device evolved as a male adaptation to sexual conﬂict over mating duration. Instead, its main function is likely to ease initiation of copulation and to further stabilize the female position throughout mating in order to resist external disturbances (in some species of Boreus, especially European, females usually do not attempt to resist and remain motionless throughout the entire copulation process which may last for hours) (Mickoleit and Mickoleit, 1976; Beutel and Friedrich, 2019). The same may be true for at least some Panorpidae (Kock et al., 2009) so this context may be relevant for extinct scorpionflies too. Please present the new findings in the context of the present literature so that the general *eLife* reader can easily follow and appreciate the broader context framing the new findings.

11) Please consider for your discussion that Zhang et al., (2021) (DOI: 10.1186/s12862-021-01771-3) reported striking swollen tarsal segments in males of the Middle Jurassic scorpionflies and hypothesized how they may have been associated with nuptial gifting behavior and served to "camouflage a gift in the mating process". These tarsal swellings have been previously known in males of some Late Jurassic Orthophlebiidae (including Orthophlebia longicauda, the one with paired anal horns), and were initially interpreted as possible pheromone glands, tympanal organs (possibly related to premating behavior), or structures for grasping female mates (Novokshonov, 1996; Willmann and Novokschonov, 1998). Either of these assumptions suggests that courtship rituals and nuptial gifting might have already been common among Jurassic scorpionflies. Consider briefly mentioning or discussing this speculative hypothesis so the general reader is aware of the ongoing debate, clarifying how the current fossils relate to the earlier work and may confirm or modify earlier interpretations.

Line items

12) Page 3, Line 16-30: Please clarify the abstract to avoid ambiguity in word meaning / interpretation by the reader. The manuscript should be clear for *eLife*'s broad readership while remaining sufficiently scientifically precise.

13) Page 3, Line 21: Well-preserved fossils are rare, meaningful ones are even more so, consider writing "well-preserved" instead.

14) Page 4, Line 28: Which group, Mecoptera or Panorpoidea, please clarify.

15) Page 4, Line 41-44: please support with references.

16) Page 5, Line 68-71: Revise into shorter clearer sentences. (e.g. we appreciate that: A good correlation has been found in extant panorpids, between their placement of nuptial-coercive mating gradient and morphology of male salivary glands and diverse non-genitalic clamping structures, such as notal-postnotal organs and anal horns. Communicating this clearly is key to engage the reader; revising using shorter sentences helps clarify further)

17) Page 7, Line 109-110: Usually, the name of a family is in Latin or Greek, hence we suggest deleting"-in Spanish.".

18) Page 10, Line 158, 159: Is there a preservation problem? Please better clarify the issue with the forewings.

19) Page 12, Line 210: strongly or reduced?

19) Page 15, Line 236: This manuscript is not about the scientific underpinnings of COVID-19, hence any reference to it has no scientific merit. Please do not name the species after the pandemic and choose a scientifically more appropriate and conventional name instead. (As a side note COVID-19 started in 2019 and not 2020 as stated by the authors: further underscoring the species naming and its motivation can be improved to meet higher scientific standards.)

20) Page 18, Line 297-299: It would help to introduce the explanation earlier at the start of this paragraph.

21) Page 18, Line 307-311: This sentence is too long and unclear.

22) Page 21, Line 366-411: Please simplify this paragraph so it can be understood by the general *eLife* reader: short and clear sentences that do not overly rely on jargon are optimal.

23) Page 24, Line 418-421: Consider moving this sentence to the beginning of this paragraph and simplify the read for a general reader.

24) Page 25, Line 450-458: There are at least two ways to make a clear comparison: one option is to use a figure that integrates all the information visually side by side; another is to use tables that organize key comparative items in a fashion that makes the comparison comprehensive and intuitive. Or both to more thoroughly substantiate the point.

25) Page 27, Line 503: Please start with a new paragraph. Structuring and organizing the different points in clear paragraphs with a well-defined subject and scope improves the read

26) Page 27, Line 512: To better communicate the evolutionary history, include a timeline in the analysis to frame the corresponding tables / plates to intuitively reflect the evolution process of different taxa and morphology.

27) Figure 1: Please note about 5% of people are color blind, so using red symbols on a green background is confusing at least 5% of the readers. Please improve the color choice, a simple white face color would provide good contrast, also increase symbol size somewhat so it stands out better. Using a white font for "SPAIN" and "MYANMAR" as well as the scale bar will improve readability since black does not have as much contrast with a dark colored background as white.

28) Figure 2: B and G do not have the same color tone as the other photos. Is F enlarged from G? Please help the reader understand and graphically show all relationships when panels represent enlarged sections specifically.

29) Figure 3: are (A) and (B) right or left wings? Please add the anatomical perspective definition for the general reader.

30) Figure 4: (C) and (D), right or left wings?

31) Figure 5: Figure A is this a photo of a complete specimen? Is C an enlarged from B? If so, please mark the specific position.

32) All figures: Please clarify the relationships between panels in figures explicitly in the caption whenever there are relationships such as enlargements, different perspectives, etcetera.

33) Figure 6: Replace "&" with "and" in full text, except references. Also unify NO and PO in the full text.

34) Figure 7: Improve the markup of the male and female in (A).

35) Supplementary Table 1: Change "Figure " to "Figure " throughout. Please add a column to indicate the age and period.

36) References: Please pay attention to the format which should be unified and strictly conform to the requirements of *eLife*.

37) ln. 181-182: "crossvein present between M4 and Cu1;" Please note that according to Figure 4C and the description (ln. 207), there are two crossveins between M4 and Cu1. Please resolve.

38) ln. 183: "anal veins strongly." is there a word missing? We only see one anal vein on the presented line drawing, are we missing something or is the description incomplete?

39) ln. 205-206: "Rs with five posterior branches; medial sector with five posterior branches"

why posterior? Does that mean there could be any anterior branches as well? Please clarify this matter in the text and response.

40) ln. 209-210: [in hindwing] "anal veins reduced". It seems that A1 is not reduced. Please consider this carefully as well as how reduction may or may not apply elsewhere.

41) ln. 324: "Orthophlebia heidemariae Willmann and Novokshonov, 1988" please correctly refer to "Willmann and Novokshonov, 1998"

42) ln. 418: "Late Jurassic Orthophlebia longicauda (†Orthophlebiidae) from Karatau (Russia)" Please note Karatau Lagerstätte is located in South Kazakhstan, not in Russia

43) ln. 428: "In a Middle Jurassic species, Orthophlebia heidemariae" this is likely erroneous, shouldn't Middle Jurassic be replaced with Late Jurassic?

44) Please consider that Burmophlebia may not be the best name for the new genus, because it sounds very similar to Burmaphlebia by Bechly et Poinar, 2013 (Odonata, Burmaphlebiidae), which will confuse both expert and general readers. Consider Burmorthophlebia as an alternative, or further improve otherwise. A thoughtful improvement would be much appreciated.

45) Please provide relevant measurements for all the morphological descriptions.

46) Regarding "male accessory gland size strongly correlates with nuptial gift size and that when male weapons are large, nuptial gifts are small and vice versa" please consider if the long gonostylus with teeth could function like a weapon or to grasp the female or win a competition among male rivals. E.g. Liu, X., Hayashi, F., Lavine, L. C., and Yang, D. (2015). Is the diversification in male reproductive traits driven by evolutionary trade-offs between weapons and nuptial gifts? Proceedings of the Royal Society B: Biological Sciences, 282(1807), 20150247. In case the authors consider this less relevant, please clarify so the reviewers can fully evaluate the authors perspective. In case the authors see merit in this perspective, please integrate it in a fashion that serves both the general and specialist reader.

47) In Figure 3 regarding the wording "restauration", use "reconstruction".

48) In the first description the justification for a new family name is provided considering differences with the next related family. However, for the other descriptions of Burmophlebia and B. pandemica [please change name] such clarifying context is missing. The differences can be found in the discussion, but systematically including them in a summarized fashion in the descriptive section helps the reader structure the findings.

49) lines 308 and 312 Please specify how long is "short", "medium-size" and "long"? Please provide all key measures explicitly using accepted length scales (SI) as requested in an earlier comment. Then also introduce the relative measures explicitly with length ranges so any reader can follow and understand the point made.

50) In addition to citing Szwedo et al., (2020) consider adding Houg et al., (2020) in case you agree this improves the literature representation. Would it be valuable to include that Haug, J.T., Azar, D., Ross, A. et al., comment on the letter of the Society of Vertebrate Paleontology (SVP) dated April 21, 2020 regarding "Fossils from conflict zones and reproducibility of fossil-based scientific data": Myanmar amber. PalZ 94, 431-437 (2020). https://doi.org/10.1007/s12542-020-00524-9. In case you do not find these citations essential, please briefly explain why so the reviewing editor can judge (*eLife* does not require additional citations unless they clearly improve the scientific representation of the literature)

51) Consider if this more recent work on Baltic amber age may help the reader frame the current findings: Kasiński, J. R., Kramarska, R., Słodkowska, B., Sivkov, V. and Piwockl, M. Paleocene and Eocene deposits on the eastern margin of the Gulf of Gdańsk (Yantarny P-1 bore hole, Kaliningrad region, Russia). Geol. Q. 64, 20-53 (2020). They concluded the age ranges between ~34 and 38 Ma.*Reviewer #1:*

This work provides new data for studying the origin and evolution of Panorpidea (Mecoptera), especially important to the exploring of sexual conflict and distinct mating mode. All data provides a critical step and deep-time perspective to illustrate the developmental reasons of gift giving in Mecoptera, and relationship between nuptial gifts and coercive behavior in the most general way.

The paper does have strengths in data, however, these strengths are not well demonstrated. In other words, without insufficiently logical analyses support the key claims, some parts are complex and make it difficult for readers to understand.

In particular:

1) About the relationship between gift-giving and coercive behavior. Most insects do not have nuptial gifts during the mating, coercive behavior is very common, which can not be a reason for Mecoptera to be models of evolutionary research. Practically, the emergence of nuptial gifts greatly improved the diversity of reproductive behavior in scorpionflies, and all different coercive behavior still at the lower level of reproduction. Therefore, nuptial gifts, rather than coercion, is the important reason for scorpionfly become an important reproductive evolutionary model.

2) In Abstract part, the most important points or conclusions are not highlighted, how many mating styles there are not the most important, but new finding provide irreplaceable evidence for the evolution of reproductive behavior in Mecoptera, or even in the Holometabola. In addition, key words should not repeat with the title.

3) In Introduction part, the extant species of scorpionflies is over 700, please pay attention to the numbers. Generally, these scorpionflies that provide the saliva globes do not provide the dead arthropods at the same time, which means usually only offering one gift at one time.

4) In Description part, should not repeat with diagnosis. Remarks of new family (Cantabridae) should compare with other families of Panorpidea, not only the Panorpidae. In addition, better compare with the Holcorpidae, as they have quite similar elongated abdomen.

5) In Discussion part, need to add some subtitles, as this part is quite long, not easy to be logical and clear. For example, Notal and postnotal organs; Anal horns; Other male abdominal processes; Male abdominal configurations and mating strategies. There are both coercive behavior and nuptial gifting in some groups, what's the reason? All groups with weak system have mixed strategy or only orthophlebiid? Need more explanation or examples. Furthermore, females have a great deal of choice in mating, and can choose to accept more than one male, based on specific behaviors and structures. Since this study lack of phylogenetic analysis, better delete the discussion on phylogenetic relationships. Of course, providing a phylogenetic analysis of particular male abdomen structures would greatly enhance the logic and comprehension of this study.*Reviewer #2:*

The authors describe new interesting scorpionfly fossils preserved in various Cretaceous and Eocene ambers, with a particular focus on certain striking morphological structures that include notal and postnotal organs and other non-genital abdominal processes. Previously, only a few examples of such structures have been known in fossil scorpionflies and they have never been treated in terms of their possible function and role in mating behavior. Based on comparison with various extant panorpid species, the authors speculate about the possible mating strategies in each new taxon under discussion in context of the evolution of mating-related sexual conflict in Mecoptera and insects in general.

The manuscript is well designed, clearly written and well structured, I enjoyed reading it. The discussion is compelling and worth publication. The descriptions of the new taxa are correct, but not without some flaws, which would be easy to fix.

However, there are two important issues that are missing in the manuscript. I think they should be considered before the manuscript is accepted for publication.

1) The mating behavior of Boreidae is not discussed, although it is well-described in literature (the paper is entitled "Evolution of sexual conflict in scorpionflies", and Boreidae are scorpionflies, aren't they?). The only mention of this family is when the authors state that "notal and postnotal organs …can be vestigially present in Boreidae", which is technically incorrect. The males of Boreidae have a non-homologous but functionally similar female-clamping device, which include hook-like rudimentary wings and structures on the 2nd and 3rd abdominal segments (vs on 4th and 5th in panorpids), usually referred to as tergal apophyses (e.g. Mickoleit and Mickoleit, 1976), that can be well-developed in some species. However, it is not apparent that this well-developed (better than in most panorpids) clamping device evolved as a male adaptation to sexual conﬂict over mating duration. Instead, its main function is likely to ease initiation of copulation and to further stabilize the female position throughout mating in order to resist external disturbances (in some species of Boreus, especially European, females usually do not attempt to resist and remain motionless throughout the entire copulation process which may last for hours) (Mickoleit and Mickoleit, 1976; Beutel and Friedrich, 2019). The same may be true for at least some Panorpidae (Kock et al., 2009), and – who knows? – maybe for some extinct scorpionflies, too.

2) There are hundreds, if not thousands, perfectly preserved and very diverse fossil scorpionflies (mainly orthophlebiids sensu lato) from the famous Mesozoic Lagerstätten in China (Middle Jurassic Daohugou Biota, Early Cretaceous Jehol Biota) available in many institutional and museum collections. I know that some of the authors have extensively studied this material. So, when the authors say that "in older fossil panorpoids, the abdomen is lacking or this portion is obscured", this is not entirely true.

Recently, Zhang et al., (2021) (DOI: 10.1186/s12862-021-01771-3) reported striking swollen tarsal segments in males of the Middle Jurassic scorpionflies and proposed a very interesting hypothesis that those might have been associated with nuptial gifting behavior and served to “camouflage a gift in the mating process”. These tarsal swellings have been previously known in males of some Late Jurassic Orthophlebiidae (including Orthophlebia longicauda, the one with paired anal horns), and were initially interpreted as possible pheromone glands, tympanal organs (possibly related to premating behavior), or structures for grasping female mates (Novokshonov, 1996; Willmann and Novokschonov, 1998). Either of these assumptions suggests that courtship rituals and nuptial gifting might have already been common among Jurassic scorpionflies. As speculative as this hypothesis may be, it should be considered or at least mentioned.*Reviewer #3:*

Nice and significant contribution about the sexual conflict mating habits of scorpionflies with hifh quality description of three new fossil panorpoid scorpionfly males from different sources of ambers from different ages. The Baltic fossil scorpionflie possesses extreme female clamping abdominal specializations, suggesting the greatest degree of sexual coercion in the group. Based on their different abdominal configurations the fossils show a wide array of mating-related morphological specializations reflecting diversified mating strategies and behaviours.

*Reviewer #4:*

Soszynska-Maj et al., explain the evolution of sexual conflict by describing three new species of Scorpionflies (Mecoptera) from three different amber resources of different ages and from different geographical areas. Surprising each of the three specimens represent a different mating behavior within the spectrum known nowadays. However, some of the fossil structures used for mating practices are more diverse and specialized, highlighting the importance of the study of fossils. The manuscript is well written and well organized. The data is nicely presented and informative. For non- scorpionflies specialists, the crucial characters are explained, also with the help of figures.

The manuscript focuses on scorpionflies for obvious reasons. First, the three specimens in amber are exceptionally well preserved, and because of their rarity in the fossil record, the specimens are of high scientific value. Second, Mecopterans are used as a model in sexual conflict studies already since many generations. Thus, the opportunity to integrate fossil specimens in such studies is crucial to understand its evolution. I considered the topic highly interesting and important not only for studying the evolution of Mecopterans but also in general for studying the evolution of insects.

Because the topic is treated seldom in the fossil record, the interest in the manuscript for other paleoentomologists would increase with a short paragraph in the introduction that includes other insects that follow similar strategies (Tettigoniidae, Megaloptera) together with some of the most relevant and general references. The species described from the Spanish amber is a new family and comparisons with recent relatives is extreme difficult. May it be necessary to look the strategies of other insects with sexual conflicts.

---

## [Author Response]

Essential revisions:General pointsThe following revisions improve both the scientific presentation and clarity for eLife's broad readership. Especially considering the paper has strengths in its unique fossil data, however, there is a need to more clearly and logically support the main points informed by the data. Several sections are unnecessary complex, which makes this manuscript unclear for both specialists and general eLife readers alike. Please address the issues thoughtfully and holistically keeping in mind eLife serves the entire biology community by publishing exceptionally clear and rigorously supported manuscripts. Any jargon deemed essential in the manuscript should be introduced in a more accessible fashion upon first use so a general reader can follow.

We have done our best to simplify the Introduction and Discussion parts, either by translating terms or simplifying the text by removing some terms. The discussion is now divided into subsections.

1) Not only the general readership needs improved higher level framing to enable a clear read. Fellow paleoentomologists would also greatly benefit from a paragraph in the introduction that includes other insects that follow similar strategies (e.g. Tettigoniidae, Megaloptera) together with some of the most relevant and general references.

The introduction is enriched by the wider context of mating behaviour, mainly among insects. However, the literature on this subject is so wide that we have limited ourselves to selected examples.

Considering the species found in Spanish amber belong to a new family, it will be challenging to make comparisons with recent relatives.

It is done.

Hence it may serve both the general and specialist reader to present the current findings in the context of mating strategies of other insects with sexual conflicts. To help introduce the general reader to the focus of this study, it would help to present an introductory figure showing this broader context, both in extant and extinct species, highlighting the unique contribution the new fossils make. Optimizing the graphic presentation and introducing key specialist terms in a clear fashion. To find examples for what a general reader would find clear consider how an in-depth National Geographic or New York Times Science article would communicate the context of the new finding and fossil as inspiration for designing an introductory figure. Given the high graphic design quality of the authors other figures, this may be relatively straightforward to add, it will greatly benefit all readers.

We looked at other articles in *eLife* on similar topics and decided to change the order of figures, to enrich them with clearer elements, such as the whole silhouette of a male and to give a drawing at the beginning to show what the article is about.

In this introductory figure and the associated introductory text, please consider there are numerous well preserved and diverse fossil scorpionflies (e.g. orthophlebiids sensu lato) from the Mesozoic Lagerstätten in China (Middle Jurassic Daohugou Biota, Early Cretaceous Jehol Biota) that are curated across institutions and museums. These collections have been studied in detail. Introducing this fact to the reader and presenting the new fossils in this established context is essential to clearly frame the new findings and show the scientific significance. In this regard the statement "in older fossil panorpoids, the abdomen is lacking or this portion is obscured" should be updated and clarified to avoid overstatement.

Comments on "in older fossil panorpoids, the abdomen is lacking or this portion is obscured": In this sentence about older material we particularly meant the Permian and Triassic Mecoptera. However, due to perhaps too much mental shortcutting, this sentence has been deleted.

2) Regarding the relationship between gift-giving and coercive behavior. Most insects do not have nuptial gifts during the mating, coercive behavior is very common, please integrate this in the framing of Mecoptera as models of evolutionary research. The emergence of nuptial gifts seems to have greatly improved the diversity of reproductive behavior in scorpionflies. Considering this, coercive behavior may perhaps still be considered at the lower level of reproduction. In this light nuptial gifts, rather than coercion, may make scorpionflies an important reproductive evolutionary model. Given the discussion of this matter by the reviewers, it is helpful to clarify the manuscript and either integrate this perspective or provide a compelling rebuttal that the reviewers can evaluate to consider the authors preferred perspective.

Both types of behaviour, a violent behaviour as well as various types of wedding gifts are widespread among animals. A spectrum of examples is now presented in the new introduction. Regarding the question which type was first in Mecoptera (at lower level of reproduction), the data presented here, as well as opinion of other authors (Zhong et al., 2015) suggest rather the reverse, i.e, that plesiomorphic (older, primary) behaviour was rather based on nuptial gifts than on coercion. The impressive armature used by males practising only coercion needed time to evolve. Indeed, is not known in fossil record older than Eocene. We added comment on this to the discussion and the abstract.

3) The abstract could more clearly communicate the key findings and conclusions. Consider further highlighting the new evidence presented for the evolution of reproductive behavior in Mecoptera (/ the Holometabola). Please revise in a fashion that serves the general eLife reader.

It was done.

4) The introduction should more carefully consider the number of extant species of scorpionflies, which is over 700.

Corrected

Consider that these scorpionflies providing saliva globes typically do not provide the dead arthropods at the same time. This has ramifications for the number of gifts that can be given at one time.

We found information that eating dead arthropods by male is necessary to produce salivary gifts, and that the male makes choice: offer a prey, or eat it itself? This would mean that at least some species can both provide either prey, or a salivary glob (Thornhill, 1990) – Introduction 117-119.

5) The descriptive and Discussion sections should be separated more clearly so the reader can digest the evidence before considering the authors evolutionary interpretation and hypothesis.

This is done now.

6) Present the new family, Cantabridae, in the context of other families of Panorpidea beyond Panorpidae. Consider comparing them further with the Holcorpidae based on their similarity in elongated abdomen. At a higher level, provide higher level context for the general eLife reader so they can follow the merit and relevance of considering the Cantabridae and contrasting them with other families through the lens of evolutionary diversity.

The comparison is now included in the remarks to the diagnosis of a new family.

7) The discussion benefits from more clearly structuring the sections with informative subtitles that help the general eLife reader grasp the big picture points. Examples of dry and challenging concepts discussed include, Notal and postnotal organs; Anal horns; Other male abdominal processes; Male abdominal configurations and mating strategies.

The discussion has been divided into parts, subchapters.

8) Please better discuss why there may be both coercive behavior and nuptial gifting in some groups, what's the functional / evolutionary reason? Do more groups have a mixed strategy or only the orthophlebiid? Thoughtful explanation supported by key examples will help clarify this issue in the discussion.

It was added.

9) Consider that females have several mating options and can choose to accept more than one male, informed by specific behaviors and morphology.

This is true, but the research on behaviour of females is based on recent species only, for obvious reasons. Therefore, we focus on the strategies of fossilized males, whose armature stands up the comparison with recent species. However, it is more than possible that is was also females’ response to extreme coercion in Baltipanorpa opressiva and damzeni that resulted in extinction of this genus. The sentence on this is added in the Conclusions, line 515-522. Also in the Introduction, line 96-100 the sentence on female’s response is added.

9) Given this study lacks a full-fledged phylogenetic analysis, consider deleting the discussion on phylogenetic relationships. Alternatively, consider providing a phylogenetic analysis of male abdomen morphology to strengthen the rigor of the study. Or if this is infeasible, but the authors wish to make a phylogenetic point, keep in mind that the only acceptable location for speculation is in the last paragraph of the discussion proposing future work in a few concise sentences (at most).

Considerations of a phylogenetic nature have been removed.

10) The mating behavior of Boreidae scorpionflies is well-described in literature, however, this context is not well-introduced and discussed. E.g. "notal and postnotal organs …can be vestigially present in Boreidae" is technically incorrect. The males of Boreidae have a non-homologous but functionally similar female-clamping device, which include hook-like rudimentary wings and structures on the 2nd and 3rd abdominal segments (vs on 4th and 5th in panorpids), usually referred to as tergal apophyses (e.g. Mickoleit and Mickoleit, 1976), that can be well-developed in some species. However, it is not apparent that this well-developed (better than in most panorpids) clamping device evolved as a male adaptation to sexual conﬂict over mating duration. Instead, its main function is likely to ease initiation of copulation and to further stabilize the female position throughout mating in order to resist external disturbances (in some species of Boreus, especially European, females usually do not attempt to resist and remain motionless throughout the entire copulation process which may last for hours) (Mickoleit and Mickoleit, 1976; Beutel and Friedrich, 2019).

You are right that, indeed, our bringing up the example of Boreidae here makes no sense, as there are their modified retracted wings which are used to hold the female. The processes on II and III tergites are not holomogous with notal and postnotal organs. After some thought Boreidae were removed from the discussion.

The same may be true for at least some Panorpidae (Kock et al., 2009) so this context may be relevant for extinct scorpionflies too. Please present the new findings in the context of the present literature so that the general eLife reader can easily follow and appreciate the broader context framing the new findings.

It has been added to the discussion.

11) Please consider for your discussion that Zhang et al., (2021) (DOI: 10.1186/s12862-021-01771-3) reported striking swollen tarsal segments in males of the Middle Jurassic scorpionflies and hypothesized how they may have been associated with nuptial gifting behavior and served to “camouflage a gift in the mating process”. These tarsal swellings have been previously known in males of some Late Jurassic Orthophlebiidae (including Orthophlebia longicauda, the one with paired anal horns), and were initially interpreted as possible pheromone glands, tympanal organs (possibly related to premating behavior), or structures for grasping female mates (Novokshonov, 1996; Willmann and Novokschonov, 1998). Either of these assumptions suggests that courtship rituals and nuptial gifting might have already been common among Jurassic scorpionflies. Consider briefly mentioning or discussing this speculative hypothesis so the general reader is aware of the ongoing debate, clarifying how the current fossils relate to the earlier work and may confirm or modify earlier interpretations.

The paper was quoted and the facts added to the discussion. When this paper was sent to *eLife*, the authors did not yet have access to this paper on leg organs, although the matter is well known to them from their work in Chinese collections.

Line items12) Page 3, Line 16-30: Please clarify the abstract to avoid ambiguity in word meaning / interpretation by the reader. The manuscript should be clear for eLife’s broad readership while remaining sufficiently scientifically precise.

The Abstract has been changed accordingly.

13) Page 3, Line 21: Well-preserved fossils are rare, meaningful ones are even more so, consider writing “well-preserved” instead.

It has been done.

14) Page 4, Line 28: Which group, Mecoptera or Panorpoidea, please clarify.

It has been done.

15) Page 4, Line 41-44: please support with references.

This sentence is now removed.

16) Page 5, Line 68-71: Revise into shorter clearer sentences. (e.g. we appreciate that: A good correlation has been found in extant panorpids, between their placement of nuptial-coercive mating gradient and morphology of male salivary glands and diverse non-genitalic clamping structures, such as notal-postnotal organs and anal horns. Communicating this clearly is key to engage the reader; revising using shorter sentences helps clarify further)

It has been done.

17) Page 7, Line 109-110: Usually, the name of a family is in Latin or Greek, hence we suggest deleting”-in Spanish.”.

It has been done.

18) Page 10, Line 158, 159: Is there a preservation problem? Please better clarify the issue with the forewings.

Corrected.

19) Page 12, Line 210: strongly or reduced?

Corrected.

19) Page 15, Line 236: This manuscript is not about the scientific underpinnings of COVID-19, hence any reference to it has no scientific merit. Please do not name the species after the pandemic and choose a scientifically more appropriate and conventional name instead. (As a side note COVID-19 started in 2019 and not 2020 as stated by the authors: further underscoring the species naming and its motivation can be improved to meet higher scientific standards.)

Changed.

20) Page 18, Line 297-299: It would help to introduce the explanation earlier at the start of this paragraph.

Corrected.

21) Page 18, Line 307-311: This sentence is too long and unclear.

Corrected.

22) Page 21, Line 366-411: Please simplify this paragraph so it can be understood by the general eLife reader: short and clear sentences that do not overly rely on jargon are optimal.

Corrected.

23) Page 24, Line 418-421: Consider moving this sentence to the beginning of this paragraph and simplify the read for a general reader.

Corrected.

24) Page 25, Line 450-458: There are at least two ways to make a clear comparison: one option is to use a figure that integrates all the information visually side by side; another is to use tables that organize key comparative items in a fashion that makes the comparison comprehensive and intuitive. Or both to more thoroughly substantiate the point.

The illustrations now are changed and supplemented to make them as synthetic as possible, and also the new Table 2 with summary of all fossil armature in males is added in the Supplementary materials.

25) Page 27, Line 503: Please start with a new paragraph. Structuring and organizing the different points in clear paragraphs with a well-defined subject and scope improves the read

Corrected.

26) Page 27, Line 512: To better communicate the evolutionary history, include a timeline in the analysis to frame the corresponding tables / plates to intuitively reflect the evolution process of different taxa and morphology.

An additional figure at the end of the discussion summarising the issue has been added. We hope this will make the article easier to perceive.

27) Figure 1: Please note about 5% of people are color blind, so using red symbols on a green background is confusing at least 5% of the readers. Please improve the color choice, a simple white face color would provide good contrast, also increase symbol size somewhat so it stands out better. Using a white font for “SPAIN” and “MYANMAR” as well as the scale bar will improve readability since black does not have as much contrast with a dark colored background as white.

Corrected.

28) Figure 2: B and G do not have the same color tone as the other photos. Is F enlarged from G? Please help the reader understand and graphically show all relationships when panels represent enlarged sections specifically.

Corrected.

29) Figure 3: are (A) and (B) right or left wings? Please add the anatomical perspective definition for the general reader.

Corrected.

30) Figure 4: (C) and (D), right or left wings?

Corrected.

31) Figure 5: Figure A is this a photo of a complete specimen? Is C an enlarged from B? If so, please mark the specific position.

Corrected.

32) All figures: Please clarify the relationships between panels in figures explicitly in the caption whenever there are relationships such as enlargements, different perspectives, etcetera.

Corrected.

33) Figure 6: Replace "&" with "and" in full text, except references. Also unify NO and PO in the full text.

Corrected.

34) Figure 7: Improve the markup of the male and female in (A).

Corrected.

35) Supplementary Table 1: Change "fig" to "Fig.” throughout. Please add a column to indicate the age and period.

The table refers to recent species; the expression “recent” has been added to the table’s caption.

36) References: Please pay attention to the format which should be unified and strictly conform to the requirements of eLife.

Corrected.

37) ln. 181-182: "crossvein present between M4 and Cu1;" Please note that according to Figure 4C and the description (ln. 207), there are two crossveins between M4 and Cu1. Please resolve.

Corrected.

38) ln. 18“: "anal veins strongly." is there a word missing? We only see one anal vein on the presented line drawing, are we missing something or is the description incomplete?

Corrected.

39) ln. 205-20“: "Rs with five posterior branches; medial sector with five posterior branches" why posterior? Does that mean there could be any anterior branches as well? Please clarify this matter in the text and response.

Corrected.

40) ln. 209-210: [in hindwing] "anal veins reduced". It seems that A1 is not reduced. Please consider this carefully as well as how reduction may or may not apply elsewhere.

Corrected.

41) ln. 32“: "Orthophlebia heidemariae Willmann and Novokshonov, 1988" please correctly refer to "Willmann and Novokshonov, 1998"

Corrected.

42) ln. 41“: "Late Jurassic Orthophlebia longicauda (†Orthophlebiidae) from Karatau (Russia)" Please note Karatau Lagerstätte is located in South Kazakhstan, not in Russia

Corrected.

43) ln. 42“: "In a Middle Jurassic species, Orthophlebia heidemariae" this is likely erroneous, shouldn’t Middle Jurassic be replaced with Late Jurassic?

Corrected.

44) Please consider that Burmophlebia may not be the best name for the new genus, because it sounds very similar to Burmaphlebia by Bechly et Poinar, 2013 (Odonata, Burmaphlebiidae), which will confuse both expert and general readers. Consider Burmorthophlebia as an alternative, or further improve otherwise. A thoughtful improvement would be much appreciated.

Thank you for this remark. Corrected.

45) Please provide relevant measurements for all the morphological descriptions.

Corrected in places where it was possible.

46) Regarding "male accessory gland size strongly correlates with nuptial gift size and that when male weapons are large, nuptial gifts are small and vice versa" please consider if the long gonostylus with teeth could function like a weapon or to grasp the female or win a competition among male rivals. E.g. Liu, X., Hayashi, F., Lavine, L. C., and Yang, D. (2015). Is the diversification in male reproductive traits driven by evolutionary trade-offs between weapons and nuptial gifts? Proceedings of the Royal Society B: Biological Sciences, 282(1807), 20150247. In case the authors consider this less relevant, please clarify so the reviewers can fully evaluate the authors perspective. In case the authors see merit in this perspective, please integrate it in a fashion that serves both the general and specialist reader.

The paper you refer to deals with Corydalidae (Megaloptera), and is known to us and cited in our paper several times. The trade-off game between weapons and nuptial gifts also occurs in Mecoptera, of course, and we refer in the revised version to this game and its role for evolution of the group in the introduction and in the discussion.

47) In Figure 3 regarding the wording "restoration", use "reconstruction".

Corrected.

48) In the first description the justification for a new family name is provided considering differences with the next related family. However, for the other descriptions of Burmophlebia and B. pandemica [please change name] such clarifying context is missing. The differences can be found in the discussion, but systematicallyincluding them in a summarized fashion in the descriptive section helps the reader structure the findings.

The name “pandemica” is changed to “opressiva”.

Differentiating diagnosis for Burmorthophlebia and Baltipanorpa opressiva are provided, as well as remarks for Burmorthophlebia.

49) lines 308 and 312 Please specify how long is "short“, "medium-size" and "long"? Please provide all key measures explicitly using accepted length scales (SI) as requested in an earlier comment. Then also introduce the relative measures explicitly with length ranges so any reader can follow and understand the point made.

Explanations have been added to the text.

50) In addition to citing Szwedo et al., (2020) consider adding Houg et al., (2020) in case you agree this improves the literature representation. Would it be valuable to include that Haug, J.T., Azar, D., Ross, A. et al., comment on the letter of the Society of Vertebrate Paleontology (SVP) dated April 21, 2020 regarding “Fossils from conflict zones and reproducibility of fossil-based scientific data”: Myanmar amber. PalZ 94, 431-437 (2020). https://doi.org/10.1007/s12542-020-00524-9. In case you do not find these citations essential, please briefly explain why so the reviewing editor can judge (eLife does not require additional citations unless they clearly improve the scientific representation of the literature)

Paper has been added.

51) Consider if this more recent work on Baltic amber age may help the reader frame the current findings: Kasiński, J. R., Kramarska, R., Słodkowska, B., Sivkov, V. and Piwockl, M. Paleocene and Eocene deposits on the eastern margin of the Gulf of Gdańsk (Yantarny P-1 bore hole, Kaliningrad region, Russia). Geol. Q. 64, 20-53 (2020). They concluded the age ranges between ~34 and 38 Ma.

Corrected.